# PI(3,4,5)P3 allosteric regulation of repressor activator protein 1 controls antigenic variation in trypanosomes

**Abdoulie O Touray**[1,2], **Rishi Rajesh**[1], **Tony Isebe**[1], **Tamara Sternlieb**[1], **Mira Loock**[1], **Oksana Kutova**[1], **Igor Cestari**[1,2]*

[1]Institute of Parasitology, McGill University, Sainte-Anne-de-Bellevue, Montreal, Canada; [2]Division of Experimental Medicine, Department of Medicine, McGill University, Montreal, Canada

**\*For correspondence:**
igor.cestari@mcgill.ca

**Competing interest:** The authors declare that no competing interests exist.

**Abstract** African trypanosomes evade host immune clearance by antigenic variation, causing persistent infections in humans and animals. These parasites express a homogeneous surface coat of variant surface glycoproteins (VSGs). They transcribe one out of hundreds of VSG genes at a time from telomeric expression sites (ESs) and periodically change the VSG expressed by transcriptional switching or recombination. The mechanisms underlying the control of VSG switching and its developmental silencing remain elusive. We report that telomeric ES activation and silencing entail an on/off genetic switch controlled by a nuclear phosphoinositide signaling system. This system includes a nuclear phosphatidylinositol 5-phosphatase (PIP5Pase), its substrate PI(3,4,5)P3, and the repressor-activator protein 1 (RAP1). RAP1 binds to ES sequences flanking VSG genes via its DNA binding domains and represses VSG transcription. In contrast, PI(3,4,5)P3 binds to the N-terminus of RAP1 and controls its DNA binding activity. Transient inactivation of PIP5Pase results in the accumulation of nuclear PI(3,4,5)P3, which binds RAP1 and displaces it from ESs, activating transcription of silent ESs and VSG switching. The system is also required for the developmental silencing of VSG genes. The data provides a mechanism controlling reversible telomere silencing essential for the periodic switching in VSG expression and its developmental regulation.

## eLife assessment

*Trypanosoma brucei* evades mammalian humoral immunity through the expression of different variant surface glycoprotein genes. In this **fundamental** paper, the authors extend previous observations that TbRAP1 both interacts with PIP5Pase and binds PI(3,4,5)P3, indicating a role for PI(3,4,5)P3 binding. They therefore suggest that antigen switching might have a signal-dependent component. The evidence is mostly **compelling**, but with some caveats because tagged proteins were used.

## Introduction

Antigenic variation is a strategy of immune evasion used by many pathogens to maintain persistent infections and entails changes in surface antigens to escape host immune clearance. The single-celled protozoa *Trypanosoma brucei spp.* circulate in the mammalian host bloodstream and periodically change their variant surface glycoprotein (VSG) coat to evade antibody clearance by antigenic variation (*Cestari and Stuart, 2018*). *T. brucei* have over 2500 VSG genes and pseudogenes, primarily on subtelomeres, and ~20 telomeric expression sites (ESs), each containing a VSG gene (*Figure 1A*). Only one VSG gene is transcribed at a time from a telomeric ES. Antigenic variation occurs via

**Figure 1.** PIP5Pase activity is essential for VSG gene silencing and switching. (**A**) Diagram of bloodstream-form ESs (BES, top) and metacyclic-form ESs (MES, bottom). (**B**) RNA-seq analysis of *T. brucei* bloodstream forms comparing 24 hr exclusive expression of Mut to WT PIP5Pase. FC, fold-change. Horizontal dotted lines indicate *p*-values at 0.05 and 0.01. Vertical dotted lines, FC at twofold. Unitig genes, genes not assembled in the reference genome. (**C–D**) RNA-seq read coverage and FC (Mut vs WT) of silent BES7 (C, top), the active BES1 (C, middle), a silent MES (C, bottom), and chromosome 4 subtelomere (**D**). Heat-map in D shows RNA-seq bins per million (BPM) reads. Gray rectangles represent genes. A 99.9% reads mapping probability to the genome (mapQ >30) retained alignments to subtelomeric regions. (**E**) VSG-seq analysis of *T. brucei* bloodstream forms after temporary (24 hr) exclusive expression of Mut PIP5Pase, and re-expression for 60 hr of WT PIP5Pase. B1-B3, biological replicates. The color shows normalized read counts per million. A 3'-end conserved VSG sequence was used to capture VSG mRNAs (*Mugnier et al., 2015*). See **Supplementary file 3** for data and gene IDs of VSGs. (**F**) VSG-seq from isolated clones after PIP5Pase temporary knockdown (24 hr) followed by its re-expression (Tet -/+) and cloning for 5–7 days. Clones of non-knockdown (tet +) cells were analyzed as controls. BES1_VSG2 (Tb427_000016000), BES12_VSG (Tb427_000008000), Chr2_5 A_VSG (Tb427_000284800), Chr9_3 A_VSG (Tb427_000553800). (**G**) Western blot of V5-tagged PIP5Pase knockdown in *T. brucei* procyclic forms. The membrane was stripped and reprobed with anti-mitochondrial heat shock protein 70 (MtHSP70). (**H**) Expression analysis of ES VSG genes after knockdown of PIP5Pase in procyclic forms by real-time PCR. Data are the mean of three biological replicates.

The online version of this article includes the following figure supplement(s) for figure 1:

**Figure supplement 1.** Clonal VSG-seq to identify VSG switching.

**Figure supplement 2.** PIP5Pase is essential for procyclic forms growth and localizes in the cell nucleus.

transcriptional switching between ESs or VSG gene recombination within ESs (*Cestari and Stuart, 2018*). The active VSG gene is developmentally regulated, and its expression is silenced in parasites encountered in the tsetse fly vector, for example procyclic and epimastigote forms. Epimastigotes develop into transmissible metacyclic trypomastigotes, which re-activate VSG gene expression before infecting mammals. The coordinated activation and silencing of telomeric ESs is essential for the periodic switch in VSG gene expression during antigenic variation and VSG developmental regulation; however, the mechanisms underlying the control of this process remain unknown.

The expressed VSG gene is transcribed by RNA polymerase I (RNAP I) from a compartment outside the nucleolus termed ES body (ESB) (*Navarro and Gull, 2001*). Transcription initiates in all ESs but only elongates through one ES (hereafter called active ES), resulting in VSG monogenic expression. The silencing of ES VSG genes involves its proximity to telomeres (*Horn and Cross, 1995*; *Rudenko et al., 1995*) and the telomere-associated factor repressor activator protein 1 (RAP1) (*Yang et al., 2009*). RAP1 is conserved among eukaryotes (*Yang et al., 2009*; *Baur et al., 2001*; *Myler et al., 2021*; *Kyrion et al., 1993*) and functions in telomere silencing (*Yang et al., 2009*; *Kyrion et al., 1993*), telomere end protection (*de Lange, 2005*; *Lototska et al., 2020*; *Platt et al., 2013*), and non-telomeric gene regulation in mammals and yeast (*Kyrion et al., 1993*; *Platt et al., 2013*). Histones and chromatin-modifying enzymes, such as histone methyltransferase and bromodomain proteins, also associate with ESs and contribute to their repression (*Figueiredo and Cross, 2010*; *Müller et al., 2018*; *Figueiredo et al., 2008*; *Schulz et al., 2015*). In contrast, the active ES is depleted of nucleosomes (*Figueiredo and Cross, 2010*) and enriched in proteins that facilitate its transcription, for example VSG exclusion (VEX) 1 and 2 and ES body 1 (ESB1), and processing of the highly abundant VSG mRNAs (*Glover et al., 2016*; *Faria et al., 2019*).

The mechanisms underlying the initiation or control of VSG switching remain unknown. Antigenic variation was thought to occur stochastically (*Deitsch et al., 2009*) with the DNA break and repair machinery involved in VSG recombination (*Boothroyd et al., 2009*; *Briggs et al., 2018*; *da Silva et al., 2018*; *McCulloch and Barry, 1999*). However, we showed that phosphoinositide signaling plays a role in the expression and switching of VSG genes (*Cestari and Stuart, 2015*; *Cestari et al., 2019*). This system entails the plasma membrane/cytosolic-localized phosphatidylinositol phosphate 5-kinase (PIP5K) and phospholipase C (PLC) (*Cestari and Stuart, 2015*), and the nuclear phosphatidylinositol phosphate 5-phosphatase (PIP5Pase). The PIP5Pase enzyme interacts with RAP1 (*Cestari et al., 2019*; *Cestari, 2019*), and either protein knockdown results in the transcription of all ESs simultaneously (*Yang et al., 2009*; *Cestari and Stuart, 2015*; *Cestari et al., 2019*). RAP1 and PIP5Pase interact with other nuclear proteins, including nuclear lamina, nucleic acid binding proteins, and protein kinases and phosphatases (*Cestari et al., 2019*). The involvement of phosphoinositide signaling in VSG expression and switching implied that signaling and regulatory processes have a role in antigenic variation linking cytosolic and nuclear proteins to control transcriptional and recombination mechanisms.

We report here that PIP5Pase, its substrate PI(3,4,5)P3, and its binding partner RAP1 form a genetic regulatory circuit that controls ES activation and repression. We show that RAP1 binds to silent telomeric ESs sequences flanking the VSG genes, and the binding is essential for VSG transcriptional repression. This association is regulated by PI(3,4,5)P3, which binds to the N-terminus of RAP1, acting as an allosteric regulator controlling RAP1-ES interactions. The system depends on PIP5Pase catalytic activity, which dephosphorylates PI(3,4,5)P3 and prevents its binding to RAP1. Inactivation of PIP5Pase results in PI(3,4,5)P3 and RAP1 binding, which displaces RAP1 from ESs, leading to transcription and switching of VSGs in the population. Hence, PIP5Pase activity is required for RAP1 association with silent ESs and VSG gene transcriptional control. The regulatory system is essential for VSG developmental silencing, and temporal disruption of this nuclear signaling system results in VSG switching. The data indicate a molecular mechanism by which ESs are periodically turned on and off during antigenic variation and parasite development.

## Results
### PIP5Pase activity controls VSG switching and developmental silencing

To investigate the role of PIP5Pase activity in VSG gene expression and switching, we performed gene expression analysis in *T. brucei* bloodstream forms that exclusively express a wildtype (WT) or a catalytic mutant D360A/N362A (Mut) PIP5Pase; the latter is unable to dephosphorylate PI(3,4,5)P3 (*Cestari et al., 2019*). The exclusive expression cells have the endogenous PIP5Pase alleles replaced by drug-selectable markers, a tetracycline (tet)-regulatable PIP5Pase allele introduced in the silent rDNA spacer, and a V5-tagged WT or Mut PIP5Pase allele in the constitutively expressed tubulin loci (*Cestari et al., 2019*). In the absence of tet, the cells exclusively express the WT or Mut PIP5Pase allele (*Cestari et al., 2019*). We performed RNA-seq after 24 hr exclusive expression of the WT or Mut PIP5Pase with Oxford nanopore sequencing (~500 bp reads) to distinguish the VSG genes expressed. The expression of the Mut PIP5Pase resulted in 1807 genes upregulated and 33 downregulated (*Figure 1B*, ≥2 fold change,

p-value ≤0.01, **Supplementary file 1**). Notably, all silent ES VSG genes were upregulated (**Figure 1B, C**), consistent with their transcription. In contrast, there was a~10-fold decrease in the active VSG (VSG2) and ESAG mRNAs (**Figure 1B, C**), implying decreased expression of the active ES (BES1) genes, perhaps resulting from competition among ESs for polymerase or factors required for their transcription. The remaining upregulated genes were primarily from silent subtelomeric arrays, largely VSG genes and pseudogenes (**Figure 1B and D**), indicating that PIP5Pase activity is also required for subtelomeric ES repression. Other upregulated genes included primarily retrotransposons, endo-nucleases, and RNA polymerases, but most chromosome core genes (Pol II and Pol III transcribed genes) were not significantly affected (**Figure 1B**). Re-establishing expression of WT PIP5Pase for 60 hr after its 24 hr withdrawal restored VSG monogenic expression. However, analysis of this population by VSG-seq revealed the expression of several VSGs other than the initially expressed VSG2 indicating a switch in VSG expression (**Figure 1E**). In contrast, only a few VSGs were detected in cells expressing WT PIP5Pase by VSG-seq (**Figure 1E**). To verify PIP5Pase role in VSG switching, we knocked down PIP5Pase for 24 hr (Tet -), then restored its expression (Tet +) and isolated clones by limit dilution and growth for 5–7 days (Figure S1). Analysis of isolated clones after temporary PIP5Pase knockdown (Tet -/+) confirmed VSG switching in 93 out of 94 (99%) analyzed clones (**Figure 1F**, **Figure 1—figure supplement 1**). The cells switched to express VSGs from silent ESs or subtelomeric regions, indicating switching by transcription or recombination mechanisms. Moreover, no switching was detected in 118 isolated clones from cells continuously expressing WT PIP5Pase (Tet +, **Figure 1F**). The data imply that PIP5Pase activity can control the transcription and switching of VSGs.

The active VSG gene is expressed in bloodstream forms but silenced during parasite development to the insect stage procyclic forms. To determine whether PIP5Pase is required for VSG develop-mental silencing, we generated procyclic conditional null cells expressing a tet-regulable V5-tagged PIP5Pase (**Figure 1—figure supplement 2**). Immunofluorescence analysis showed that PIP5Pase-V5 localizes in the nucleus of procyclic forms (**Figure 1—figure supplement 2**), as in bloodstream forms (**Cestari and Stuart, 2015**). Western analysis showed that PIP5Pase-V5 proteins were eliminated 72 hr after knockdown (**Figure 1G**), and growth arrest was detected after 5 days of knockdown (**Figure 1—figure supplement 2**). Gene expression analysis showed a 5–15 log2-fold upregulation of all ES VSG genes at 72 hr (**Figure 1H**) at a time when cells are viable and dividing. The data indicate that VSG switching and developmental silencing depend on PIP5Pase activity.

## RAP1 bind to sequences flanking VSG genes in silent telomeric ESs

Because PIP5Pase and its substrate PI(3,4,5)P3 interact with RAP1 (**Cestari et al., 2019**), we postu-lated that VSG silencing might involve the regulation of RAP1's repressive function. We showed that RAP1 binds telomeric or 70 bp repeats (**Cestari et al., 2019**), but it is unknown if it binds to other ES sequences or genomic loci. We performed ChIP-seq with a cell line expressing an in-situ HA-tagged RAP1 and nanopore sequencing (~500 bp reads). We found that RAP1-HA binds primarily to silent ESs at the 70 bp and telomeric repeats, which are sequences upstream and downstream of the VSG genes in bloodstream-form ESs, respectively (**Figure 2A–B and E** and **Figure 2—figure supplement 1**; p-values<$10^{-4}$). There was also a slight enrichment of RAP1-HA in the 50 bp repeat sequences preceding the ES promoters (**Figure 2A** and **Figure 2—figure supplement 1**; p-values <$10^{-4}$). However, RAP1-HA did not bind significantly to genes, including VSG genes or pseudogenes in the ESs or subtelomeric regions (**Figure 2A–B and F**). Notably, analysis of uniquely mapped reads showed that RAP1-HA was depleted from the active ES (**Figure 2C–D** and **Figure 2—figure supple-ment 2**), consistent with RAP1 having a repressive function. The RAP1-HA bound sequences are rich in TAAs and GTs, including the 70 bp $(TTA)_7(GT/A)_9$ and telomeric $(TTAGGG)_n$ repeats (**Figure 2A**), but also T and G/T-rich sequences downstream of VSG genes. RAP1-HA was also bound to metacy-clic ESs at AT/GC-rich or $(TAACCC)n$ repeat sequences near VSG genes (**Figure 2E** p-values < $10^{-4}$, **Figure 2—figure supplement 3**). RAP1-HA also bound sparsely to subtelomeric regions, including centromeres (**Figure 2F** p-values < $10^{-4}$), which are AT-rich in trypanosomes (**Obado et al., 2007**), and some binding sites overlapped with the centromeric protein kinetoplastid kinetochore 2 (KKT2) (**Akiyoshi and Gull, 2014**). Other poorly enriched RAP1-HA peaks in the subtelomeric regions reflect residual ES sequences from recombination (**Figure 2F**). Hence, RAP1-HA binds primarily to telomeric and 70 bp repeats within silent ESs. The 70 bp repeats are ES-specific sites for RAP1 binding near silent VSG genes.

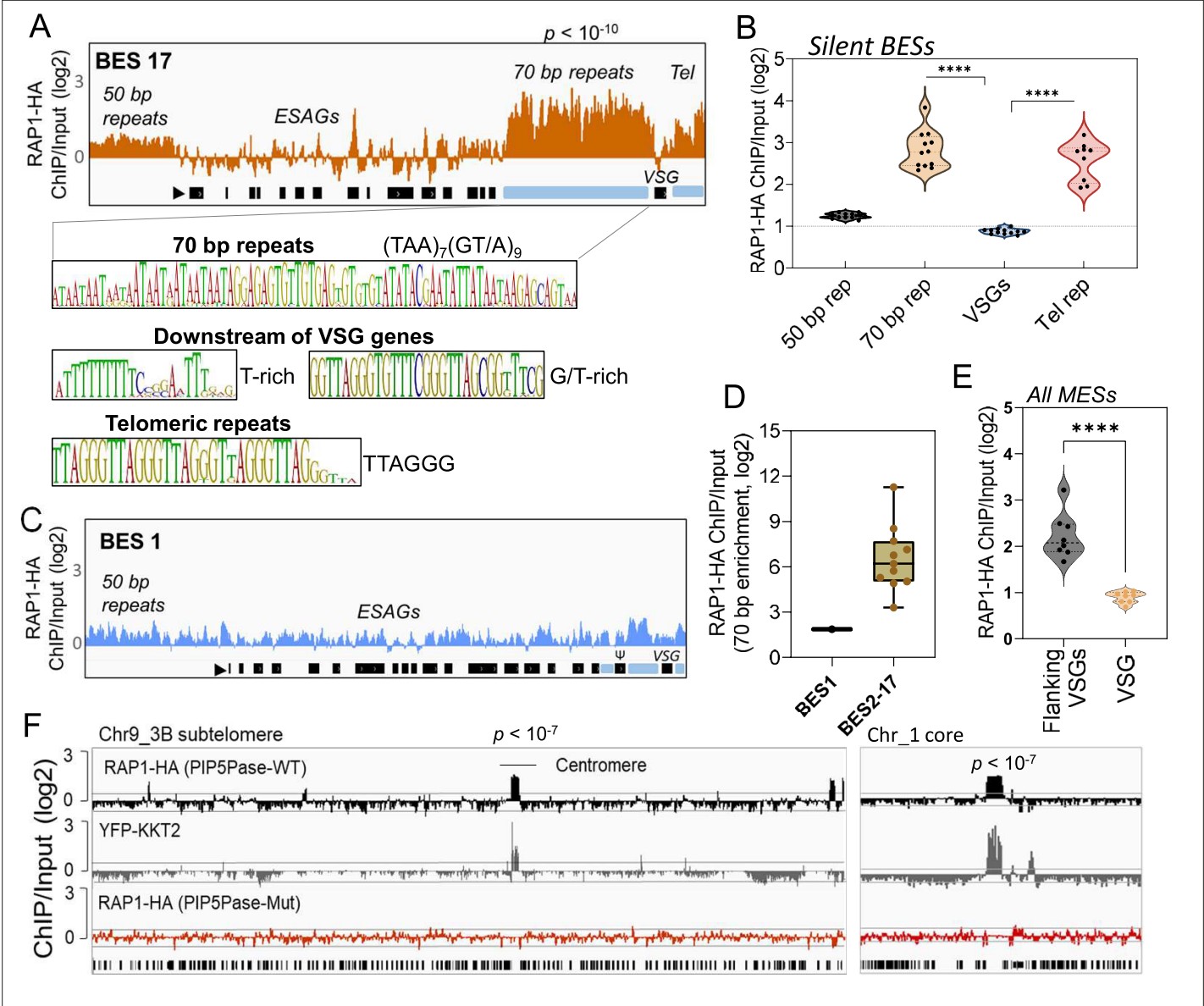

**Figure 2.** ChIP-seq analysis of RAP1-HA in *T. brucei*. (**A**) RAP1-HA binding sites on the silent BES17. Data show fold-change comparing ChIP vs Input. Below is the sequence bias of RAP1-HA bound regions. Black rectangles, ES genes; black triangle, ES promoter; cyan rectangle, 70 bp and telomeric repeats. (**B**) RAP1-HA binding to selected regions in silent BES sequences. Each dot represents the mean of a silent BES. (**C**) RAP1-HA enrichment to the active BES1 (fold-change comparing ChIP vs Input). See **Supplementary file 4** for additional read mapping and filtering analysis. (**D**) Comparison of RAP1-HA binding to 70 bp in silent and active ESs. (**E**) RAP1-HA enrichment over all MESs. Flanking VSGs, DNA sequences upstream or downstream of VSG genes in MESs. (**F**) RAP1-HA binding to subtelomere 3B of chromosome (Chr) 9 (left) or chromosome 1 core (right). Yellow fluorescent protein (YFP)-tagged KKT2 protein ChIP-seq from **Akiyoshi and Gull, 2014** is shown. RAP1-HA ChIP-seq in cells expressing Mut PIP5Pase is shown below. p-values (**p**) were calculated using Model-based Analysis of ChIP-Seq (MACS) from three biological replicates. Data show fold-change of ChIP vs Input analysis. See **Supplementary files 2 and 4** for detailed statistics. ****, p-value <0.0001 using two-tailed unpaired t-test.

The online version of this article includes the following figure supplement(s) for figure 2:

**Figure supplement 1.** RAP1-HA binding to silent telomeric ESs by ChIP-seq in cells expressing WT or Mut PIP5Pase.

**Figure supplement 2.** ChIP-seq of RAP1-HA binding to a silent (BES17) vs the active ES (BES1) after multiple alignment filtering parameters.

**Figure supplement 3.** MES Chr11_5 A data are included in the main manuscript (**Figure 5**).

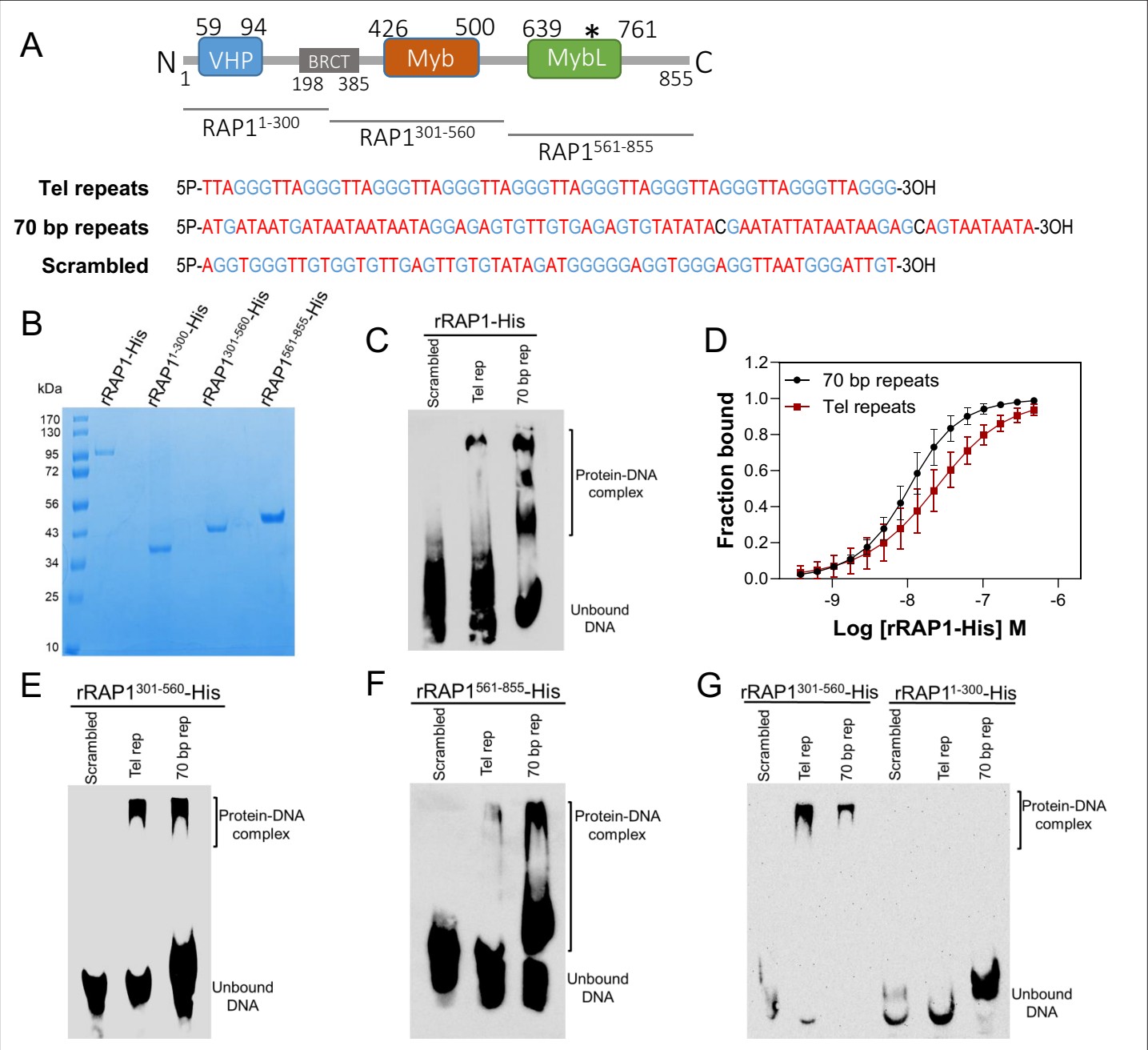

**Figure 3.** rRAP1-His binds to telomeric and 70 bp repeats via its Myb and MybL domain. (**A**) Top, RAP1 diagram shows VHP, BRCT, Myb and MybL domains. Numbers indicate residue positions; asterisk, nuclear targeting sequence. Bottom, telomeric (10 repeats), 70 bp (one repeat), or scrambled (generated from telomeric repeats) sequences used in binding assays. (**B**) His-tagged rRAP1, rRAP1$^{1-300}$, rRAP$^{301-560}$, and rRAP1$^{561-855}$ resolved in 10% SDS/PAGE and Coomassie-stained. (**C**) EMSA of His-tagged rRAP1 with biotinylated telomeric repeats (Tel rep), 70 repeats (70 bp rep), or scrambled sequences resolved in 6% native/PAGE and developed with Streptavidin-HRP. (**D**) MST binding kinetics of rRAP1-His with Cy5-labelled telomeric repeats or 70 bp repeats. Data shown are the mean ± SDM of four biological replicates. (**E–G**) EMSA of rRAP1$^{301-560}$-His (**E**), rRAP1$^{561-855}$-His (**F**), or rRAP1$^{1-300}$-His (**G**) with telomeric, 70 bp repeats, or scrambled sequences resolved in 6% native/PAGE. rRAP1$^{301-560}$-His were used as a positive control in G.

The online version of this article includes the following figure supplement(s) for figure 3:

**Figure supplement 1.** Recombinant His-tagged RAP1 (rRAP1-His) binds to 70 bp, telomeric repeats, and PI(3,4,5)P3.

**Table 1.** Binding kinetics of rRAP1 to telomeric repeats, 70 bp repeats, and phosphoinositides. Data show the mean ± standard deviation of the mean (SDM).

| Interactions | Kd [M]±SDM |
|---|---|
| rRAP1-His+Telomeric repeats | $24.1 \times 10^{-9} \pm 2.0 \times 10^{-9}$ |
| rRAP1-His+70 bp repeats | $10.0 \times 10^{-9} \pm 0.33 \times 10^{-9}$ |
| rRAP1-His+PI(3,4,5)P3 | $19.7 \times 10^{-6} \pm 2.8 \times 10^{-6}$ |
| rRAP1-His+PI(4,5)P2 | No binding |
| rRAP1-His+Telomeric repeats+PI(3,4,5)P3 | $154.6 \times 10^{-9} \pm 14.2 \times 10^{-9}$ |
| rRAP1-His+Telomeric repeats+PI(4,5)P2 | $19.4 \times 10^{-9} \pm 1.5 \times 10^{-9}$ |
| rRAP1-His+70 bp repeats+PI(3,4,5)P3 | $187.7 \times 10^{-9} \pm 29.9 \times 10^{-9}$ |
| rRAP1-His+70 bp repeats+PI(4,5)P2 | $17.5 \times 10^{-8} \pm 0.9 \times 10^{-9}$ |

To confirm RAP1 binding to ES sequences, we expressed and purified from *E. coli* recombinant 6xHis-tagged *T. brucei* RAP1 (rRAP1-His) (*Figure 3A–B*). We performed electrophoretic mobility shift assays (EMSA) using rRAP1-His and biotinylated telomeric repeats, 70 bp repeat, or a scrambled telomeric repeat sequence as a control. rRAP1-His bound to telomeric and 70 bp repeats but not to the scrambled DNA sequence (*Figure 3C*). We obtained similar results by microscale thermophoresis (MST) kinetics using rRAP1-His and Cy5-labelled DNA sequences (*Figure 3D*, *Figure 3—figure supplement 1*). The MST analysis revealed a Kd of 24.1 and 10 nM of rRAP1-His for telomeric and 70 bp repeats, respectively (*Table 1*). RAP1 has two DNA binding domains, a central Myb (position E426-Q500) and a C-terminal Myb-like (MybL, position E639-R761) domain (*Figure 3A*), and a divergent BRCT domain (S198-P385) (*Yang et al., 2009*), which is involved in RAP1 self-interaction (*Afrin et al., 2020b*) and perhaps other protein interactions (*Cestari and Stuart, 2015*; *Cestari et al., 2019*). To identify which RAP1 domain mediates DNA interactions, we performed EMSA with rRAP1$^{1-300}$-His (aa positions in superscript), rRAP1$^{301-560}$-His, and rRAP1$^{561-855}$-His, the last two fragments encompass the Myb and MybL domains, respectively (*Figure 3A–B*). Both rRAP1$^{301-560}$-His and rRAP1$^{561-855}$-His proteins bound to telomeric or 70 bp repeats (*Figure 3E–F*, *Figure 3—figure supplement 1*), but no DNA binding was detected with the N-terminal rRAP1$^{1-300}$-His (*Figure 3G*). Because all proteins were His-tagged and no binding was detected to scrambled DNA nor with rRAP1$^{1-300}$-His, unspecific DNA binding by the His-tag is ruled out. Our data show that both Myb and MybL can independently and directly bind to 70 bp and telomeric repeats.

## PIP5Pase activity controls PI(3,4,5)P3 binding to the N-terminus of RAP1

We found in the N-terminus of RAP1 a villin headpiece (VHP) domain (*Figure 3A*, position Y59 to F94, *e*-value <10$^{-4}$), which is typically present in Villin proteins and binds phosphoinositides (*Kumar et al., 2004*). *T. brucei* RAP1 VHP domain is conserved (~27% aa identity) with other VHPs in Villin proteins forming a three helices fold over a hydrophobic core (*Chiu et al., 2005*; *Meng and McKnight, 2009*; *Figure 4A*), which is essential for phosphoinositide binding (*Kumar et al., 2004*). We performed binding assays with rRAP1-His and biotinylated phosphoinositides (*Cestari, 2019*) and confirmed that rRAP1-His binds specifically to PI(3,4,5)P3 (*Figure 3—figure supplement 1*; *Cestari et al., 2019*). Moreover, we found that rRAP1$^{1-300}$-His binds to PI(3,4,5)P3 (*Figure 4B*), but not other phosphoinositides or inositol phosphates tested (*Figure 4C*) and the binding was competed by a molar excess of unlabelled PI(3,4,5)P3 (*Figure 4D*). However, no PI(3,4,5)P3 binding was detected with rRAP1$^{301-560}$-His or rRAP1$^{561-855}$-His proteins, indicating that RAP1 binds to PI(3,4,5)P3 via its N-terminus containing the VHP domain. Moreover, binding kinetics by differential scanning fluorescence with unlabelled phosphoinositides showed that rRAP1-His binds to PI(3,4,5)P3 with a Kd of 19.7 μM (*Figure 4E*, *Table 1*). To determine if RAP1 binds to PI(3,4,5)P3 in vivo, we in-situ HA-tagged RAP1 in cells that express the WT or Mut PIP5Pase and analyzed endogenous PI(3,4,5)P3 levels associated with immunoprecipitated RAP1-HA. There were ~100 fold more PI(3,4,5)P3 associated with RAP1-HA in cells expressing the Mut than the WT PIP5Pase (*Figure 4F*, *Figure 4—figure supplement 1*). In contrast, there was a mild

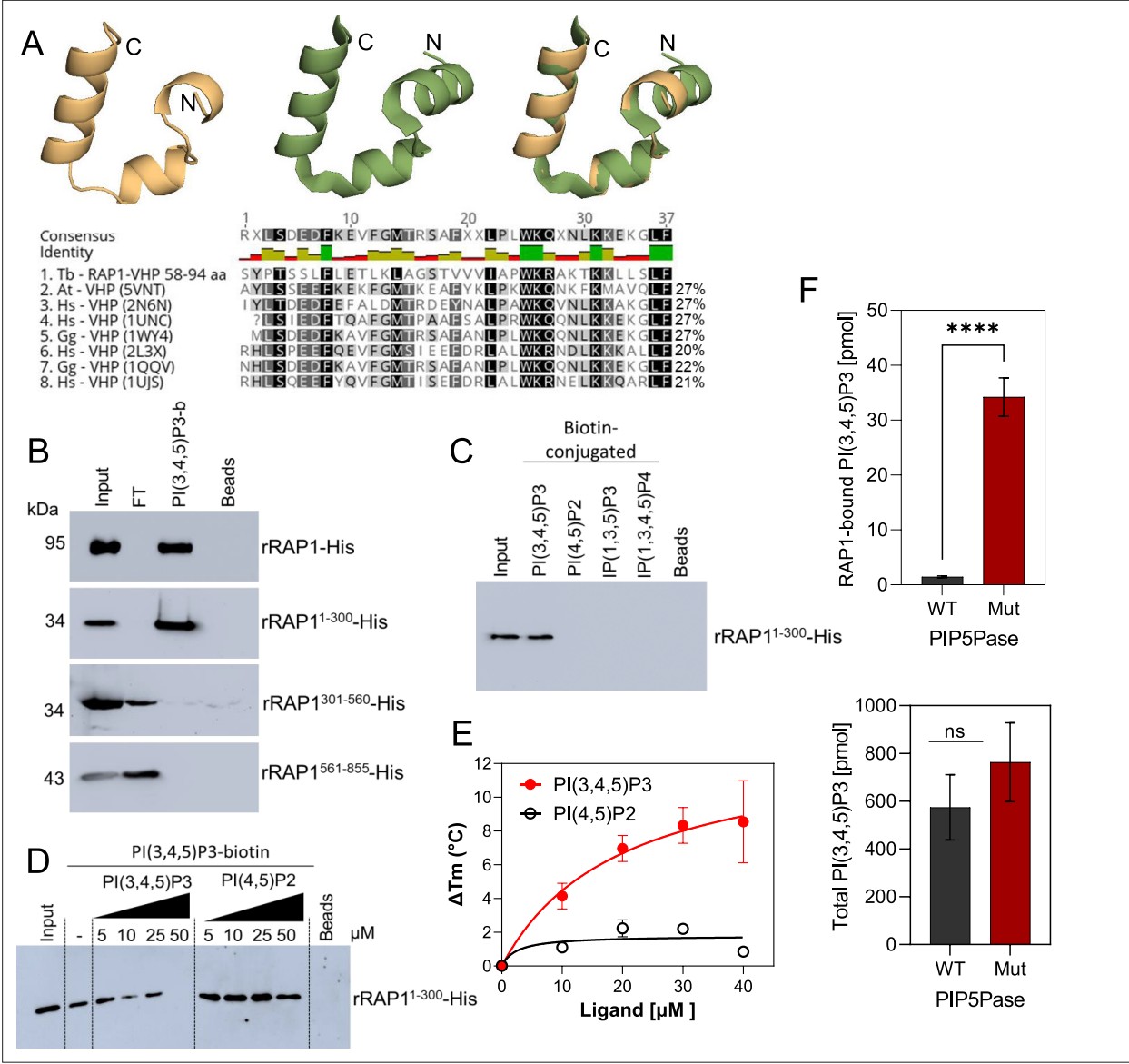

**Figure 4.** rRAP1-His binds to PI(3,4,5)P3 through its N-terminus. (**A**) Modeling and alignment of RAP1 VHP domain. *Left*, RAP1 modeled structure; *middle*, VHP domain structure of human supervillin protein (PDB accession number 2K6N); and *right*, superposition of *T. brucei* RAP1 modeled VHP and human supervilin VHP domains. Alignment of *T. brucei* RAP1 VHP with human (Hs), *Arabidopsis thaliana* (At), and *Gallus gallus* (Gg) VHP domains from Villin proteins. PDB accession numbers are indicated in parenthesis. % of aa identity to *T. brucei* sequence are shown. (**B**) Binding assays with rRAP1-His, rRAP1^1-300-His, rRAP301-560-His, or rRAP1561-855-His and PI(3,4,5)P3-biotin. Beads, Streptavidin-beads; FT, flow-through. Proteins were resolved in 10% SDS/PAGE and Western developed with α-His mAbs. (**C**) Binding of His-tagged rRAP1^1-300 to biotinylated phosphoinositides or IPs. (**D**) Binding of rRAP1^1-300-His to PI(3,4,5)P3-biotin in presence of unlabelled PI(3,4,5)P3 or PI(4,5)P2. For C and D, proteins were analyzed as in B. (**E**) Binding kinetics of rRAP1-His with unlabelled PI(3,4,5)P3 or PI(4,5)P2. ΔTm, change in melting temperature. Data show the mean ± SDM of three biological replicates. (**F**) Quantification of RAP1-bound PI(3,4,5)P3 (top) or total cellular PI(3,4,5)P3 (bottom) levels in *T. brucei* exclusively expressing WT or Mut PIP5Pase. Data show the mean ± SDM of four biological replicates. ****, p-value <0.0001 using two-tailed unpaired t-test.

The online version of this article includes the following figure supplement(s) for figure 4:

**Figure supplement 1.** Quantification of PI(3,4,5)P3 from *T. brucei* bloodstream form total cell extract or associated with RAP1-HA.

but not significant difference in the total cellular PI(3,4,5)P3 levels comparing cells expressing WT versus Mut PIP5Pase (*Figure 4F*), implying that PIP5Pase activity controls a localized pool of PI(3,4,5) P3 in the nucleus available for RAP1 binding. Hence, RAP1 binds specifically to PI(3,4,5)P3 via its N-terminus in a PIP5Pase activity-dependent fashion.

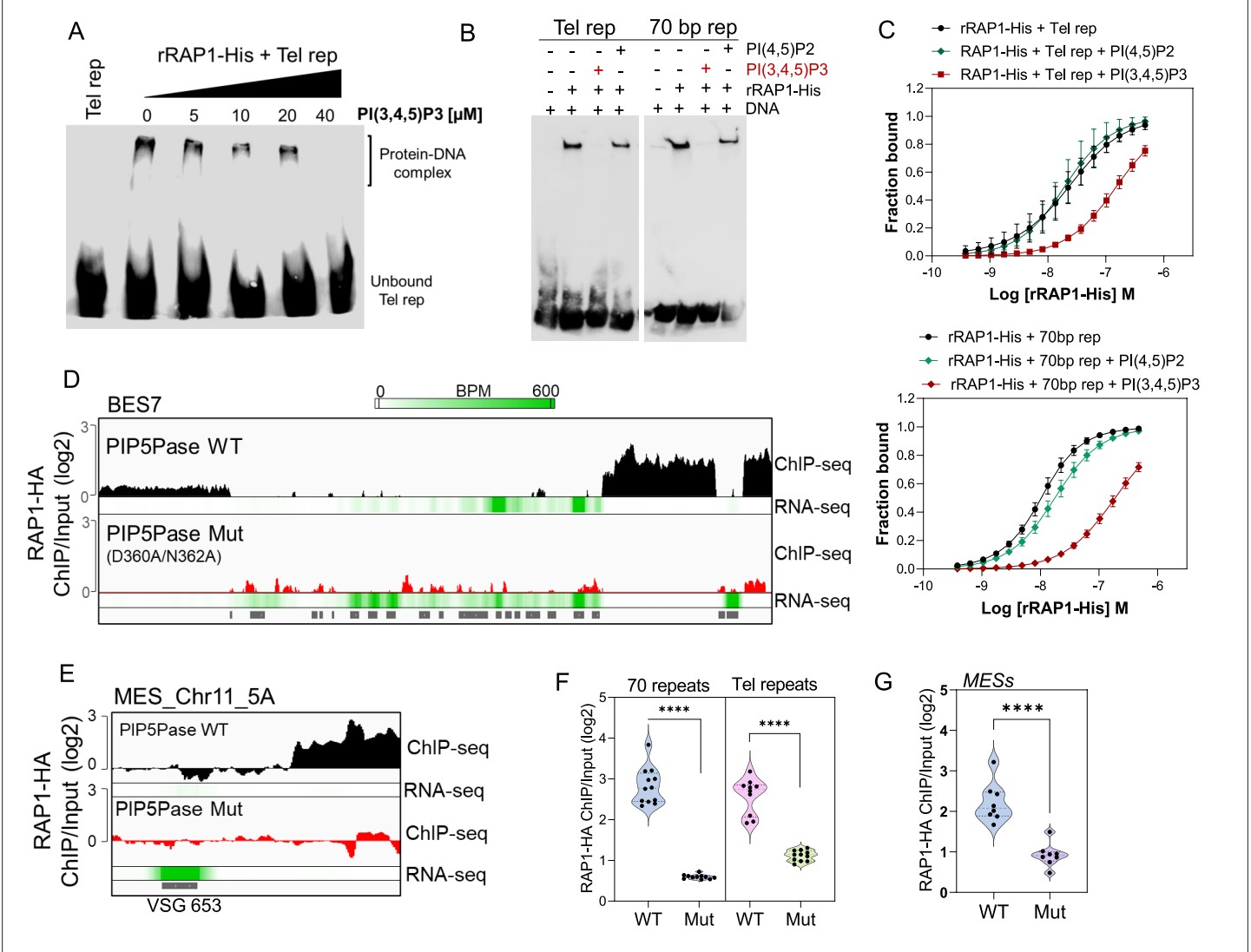

**Figure 5.** PIP5Pase controls rRAP1-His binding to telomeric ESs via PI(3,4,5)P3. (**A**) EMSA of rRAP1-His with biotinylated telomeric repeats and increasing concentrations of PI(3,4,5)P3. (**B**) EMSA of rRAP1-His with biotinylated telomeric repeats (left) or 70 bp repeats (right) and 30 µM of PI(3,4,5)P3 or PI(4,5)P2. For A-B, samples were resolved in 6% native/PAGE, transferred to nylon membranes, and developed with streptavidin-HRP. (**C**) MST binding kinetics of rRAP1-His with Cy5-labelled telomeric repeats (top) or 70 bp repeats (bottom) with 30 µM of PI(3,4,5)P3 or PI(4,5)P2. Data show the mean ± SDM of four biological replicates. (**D–E**) ChIP-seq of RAP1-HA binding to BES7 (**D**) or (MES_Chr11_5 A) (E) from cells that exclusively express WT or Mut PIP5Pase for 24 hr. RNA-seq comparing exclusive expression of Mut vs WT PIP5Pase for 24 hr is shown. (**F–G**) Violin plots show RAP1-HA mean enrichment over 70 bp or telomeric repeats from all silent BESs (**F**) or MESs (**G**). Each dot represents an ES. BPM, bin per million. ChIP-seq and RNA-seq were performed in three biological replicates. ****, p<0.0001 using two-tailed unpaired t-test.

The online version of this article includes the following figure supplement(s) for figure 5:

**Figure supplement 1.** PI(3,4,5)P3 control of VSG expression and model of VSG switching and developmental silencing.

## PI(3,4,5)P3 is an allosteric regulator of RAP1 and controls telomeric ES repression

Because RAP1 binds to PI(3,4,5)P3 via its N-terminus and to ESs via its central and C-terminus Myb and Myb-L domains, we posited that PI(3,4,5)P3 binds to RAP1 and controls its association with ESs. We performed EMSA binding assays and found that PI(3,4,5)P3 inhibited rRAP1-His binding to telomeric repeats in a dose-dependent fashion, but no inhibition was detected with PI(4,5)P2 (*Figure 5A–B*). Similar PI(3,4,5)P3 binding inhibition of rRAP1-His was observed for 70 bp repeats (*Figure 5B*). Moreover, MST binding kinetics with 30 µM of PI(3,4,5)P3 increased rRAP1-His Kd to telomeric and 70 bp repeats in ~6.5-fold and ~18.5-fold, respectively (*Figure 5C, Table 1*). Notably, PI(3,4,5)P3 did not

affect rRAP1[301-560]-His or rRAP1[561-855]-His binding to telomeric or 70 bp repeats (*Figure 3—figure supplement 1*), implying that PI(3,4,5)P3 does not compete with Myb or MybL domains direct binding to DNA. PI(3,4,5)P3 inhibition of RAP1-DNA binding might be due to its association with RAP1 N-terminus causing conformational changes that affect Myb and MybL domains association with DNA.

Our data suggest a model in which PI(3,4,5)P3 levels control RAP1 binding to ESs and thus silencing and activation of VSG genes. To evaluate this model, we performed ChIP-seq with RAP1-HA in cells that exclusively express WT or Mut PIP5Pase. The Mut PIP5Pase expression results in local PI(3,4,5) P3 available for RAP1 binding (*Figure 4F*), whereas WT PIP5Pase dephosphorylates PI(3,4,5)P3 into PI(3,4)P2, which cannot bind RAP1 (*Cestari et al., 2019*). The expression of Mut PIP5Pase abolished RAP1-HA binding to 70 bp, 50 bp, and telomeric repeats in all bloodstream and metacyclic ESs (*Figure 5D–G*). Notably, the decreased RAP1-HA binding to ESs correlates with increased expression of VSGs and ESAG genes (*Figure 5D–E*, RNAseq), indicating that PIP5Pase controls RAP1 binding to ESs and thus VSG gene expression. The increased level of VSG and ESAG mRNAs detected in cells expressing Mut PIP5Pase (*Figure 5D*) may reflect increased Pol I transcription. It is possible that the low levels of RAP1-HA at the 50 bp repeats affect Pol I accessibility to the BES promoter; alternatively, RAP1 association to telomeric or 70 bp repeats may affect chromatin compaction or folding impairing VSG and ESAG genes transcription. The expression of Mut PIP5Pase also removed RAP1-HA peaks in subtelomeric regions (*Figure 2F*), which might impact subtelomeric chromatin organization and thus VSG expression and recombination (*Figure 1D–F*). We showed that RAP1 interacts with PIP5Pase within a 0.9 MDa complex, and this association is stable in cells expressing the Mut PIP5Pase (*Cestari et al., 2019*). Hence, RAP1 dissociation from ESs is unlikely the result of PIP5Pase mutations affecting the complex integrity. Our data indicate that PIP5Pase activity regulates RAP1 binding to DNA via PI(3,4,5)P3, thus controlling the reversible silencing of telomeric VSG genes.

## Discussion

We found that the reversible silencing of telomeric VSG genes in *T. brucei* is controlled by a phosphoinositide regulatory system. The system operates via PI(3,4,5)P3 regulation of RAP1-DNA binding and is controlled by PIP5Pase enzymatic activity. The system functions as an on/off genetic switch to control telomeric ES activation and silencing and indicates a mechanism to control the periodic switching of VSG genes during antigenic variation and VSG developmental silencing. The regulation requires PIP5Pase activity for continuous ES repression via dephosphorylation of PI(3,4,5)P3. PIP5Pase temporary inactivation results in the accumulation of PI(3,4,5)P3 bound to the RAP1 N-terminus. This binding displaces RAP1 Myb and MybL domains from DNA, likely due to RAP1 conformational changes, allowing polymerase elongation through the VSG gene. PI(3,4,5)P3 acts as a typical allosteric regulator controlling RAP1-DNA interactions and, thus, VSG transcriptional repression.

Our data indicate that regulation of PIP5Pase activity and PI(3,4,5)P3 levels play a role in VSG switching. Other phosphoinositide enzymes such as PIP5K or PLC, which act on PI(3,4,5)P3 precursors, also affect VSG expression and switching (*Cestari and Stuart, 2015*), arguing that this system is part of a cell-wide signaling network and includes the transcriptional and recombination machinery. The finding of RAP1 binding to subtelomeric regions other than ESs, including centromeres, requires further validation. Nevertheless, it suggests a role for RAP1 repressing subtelomeric chromatin. Chromatin conformational capture analysis showed that subtelomeric regions and centromeres (many subtelomeric in *T. brucei*) have a high frequency of interactions forming the boundaries of genome compartments in *T. brucei* (*Müller et al., 2018*). Disrupting PIP5Pase activity affects RAP1 association with those regions and, thus, might impact long-range chromatin interactions. This organization may explain RAP1's role in VSG recombinational switching (*Nanavaty et al., 2017*) and the transcription of subtelomeric VSGs upon PIP5Pase mutation, RAP1 (*Afrin et al., 2020a*) or NUP1 knockdowns (*DuBois et al., 2012*). Although we showed that C-terminal HA-tagged RAP1 protein has telomeric localization (*Cestari and Stuart, 2015*) and interactions with other telomeric proteins (*Cestari et al., 2019*), we cannot rule out potential differences between HA-tagged and non tagged RAP1.

RAP1 and PIP5Pase interact and localize near the nuclear periphery (*Cestari and Stuart, 2015*; *Cestari et al., 2019*). RAP1 also associates with NUP1 (*Cestari et al., 2019*), which knockdown derepresses silent VSG genes (*DuBois et al., 2012*). PI(3,4,5)P3 and other phosphoinositides are synthesized in the endoplasmic reticulum (ER) and Golgi (*Martin and Smith, 2006*; *Kim et al., 2011*) and distributed to organelles, including the nuclear membrane (*Cestari and Stuart, 2015*; *Kim et al.,*

*2011*). The scenario suggests a model in which RAP1 associates with silent telomeric ESs and lamina proteins at the nuclear periphery (*Müller et al., 2018*; *Cestari and Stuart, 2015*; *DuBois et al., 2012*), where regulation of silencing and activation might occur. The compartmentalization of silent ESs near the nuclear lamina may prevent the association of factors required for ES transcription, including chromatin-associated proteins and RNA processing factors. In contrast, the active ES is depleted of RAP1, a process that might involve ES mRNAs competition for RAP1 binding to ES DNA (*Gaurav et al., 2023*), and it is enriched in proteins that facilitate transcription, including VEX2 and ESB1 (*Faria et al., 2019*; *López-Escobar et al., 2022*). Furthermore, the active ES localizes in a compartment that favours efficient processing of VSG mRNAs (*Faria et al., 2021*; *Budzak et al., 2022*). Hence, the active and silent ESs seem to occupy distinct subnuclear compartments and have specific associated proteins. Within this model, VSG switching may entail the inactivation of PIP5Pase, leading to reorganization of silent ES chromatin, for example RAP1 removal from ESs, ESs relocation away from the nuclear lamina, and perhaps changes in ES three-dimensional organization (*Figure 5—figure supplement 1*). Once RAP1 is dissociated from ESs, other factors, such as ESB1 and VEX2, may associate with the silent ESs to initiate transcription. Although the requirements for establishing VSG monogenic expression are unknown, it may entail the successful association of factors with ESs for their transcription or recruitment to the ESB (*Navarro and Gull, 2001*; *Faria et al., 2021*).

How the phosphoinositide signaling system is initiated to control VSG switching is unknown, but our data suggest an alternative non-stochastic model regulating antigenic variation. It is unknown if this signaling system regulates antigenic variation in vivo. Nevertheless, the data indicate that trypanosomes may have evolved a sophisticated mechanism to regulate antigenic variation that entails genetic and signaling processes, and such processes may be conserved in other pathogens that rely on antigenic variation for infection, for example *Plasmodium* and *Giardia*, broadening opportunities for drug discovery. The allosteric regulation of RAP1 by PI(3,4,5)P3 denotes a mechanism for phosphoinositide regulation of telomere silencing. Given that telomere silencing is conserved and dependent on RAP1 in most eukaryotes, this regulatory system may be present in other eukaryotes.

# Materials and methods
## Cell culture, cell lines, and growth curves

*T. brucei* bloodstream forms (BF) single marker 427 conditional null (CN) and V5- or HA-tagged cell lines were generated and maintained in HMI-9 at 37 °C with 5% $CO_2$ as described (*Cestari and Stuart, 2015*). *T. brucei* were cultured at McGill University under permit B-07840 (for I. Cestari laboratory). Cell lines that exclusively express wildtype (WT) or mutant D360A/N360A (Mut) PIP5Pase gene (Tb927.11.6270) were generated as previously described (*Cestari et al., 2019*). The WT or Mut PIP5Pase mRNAs exclusively expressed from tubulin loci are 1.6 and 1.3-fold the WT PIP5Pase mRNA levels expressed from endogenous alleles in the single marker 427 strain. The fold-changes were calculated from RNA-seq counts per million from this work (WT and Mut PIP5Pase, *Supplementary file 1*) and our previous RNA-seq from single marker 427 strain (*Cestari et al., 2019*). *T. brucei* 29.13 were maintained in SDM-79 medium supplemented with 10% FBS, G418 (2.5 µg/mL), and hygromycin (50 µg/mL) at 27 °C. To generate *T. brucei* procyclic forms (PF) CN PIP5Pase, PF 29.13 was transfected with a V5-tagged PIP5Pase gene cloned into pLEW100-3V5 (*Cestari and Stuart, 2015*). The plasmid was integrated into the rRNA silent spacer, and transgenic cells were selected by resistance to phleomycin (5 µg/mL). The PIP5Pase endogenous alleles were then sequentially replaced by homologous recombination with PCR constructs containing about 500 bp of the PIP5Pase 5'UTR and 3'UTR flanking a puromycin or blasticidin drug resistance gene (*Cestari and Stuart, 2015*). The CRISPR/Cas9 ribonucleoprotein complex was used to cut the second endogenous allele and increase homologous recombination rates. Guides targeting 5' and 3' regions (see primer sequences in *Supplementary file 6*) of the PIP5Pase coding sequence were produced by in vitro transcription with T7 polymerase (Promega), and co-transfected with recombinant saCas9 Nuclease NLS Protein (Applied Biological Materials Inc), and a PCR-based puromycin resistance construct containing 5'- and 3'-UTRs of the targeted gene. Note that this procedure increased the efficiency of recombination by about 20 times. The *T. brucei* 427 strain, or mutants derived from this strain, were verified by PCR analysis and DNA sequencing. Cumulative growth curves were performed by diluting *T. brucei* PFs to 2×10⁶ parasites/mL in the absence or presence of tetracycline (tet, 0.5 µg/mL). The culture growth was counted every

two days using a Coulter Counter (Beckman Coulter) for twelve days. Cell lines generated in this work are available upon request.

## Immunofluorescence and western blotting

*Immunofluorescence*: Immunofluorescence was done using *T. brucei* PF PIP5Pase CN, which expresses V5-tagged WT PIP5Pase in the presence of 0.1 μg/mL of tet. Mid-log growth phase parasites were washed three times in PBS with 6 mM glucose (PBS-G) for 10 minutes (min). Then, $2.0x10^6$ cells were fixed in 2% paraformaldehyde in PBS for 10 min at room temperature (RT) onto poly L-lysine treated 12 mm glass coverslips (Fisher Scientific). The coverslips were washed three times in PBS, and cells permeabilized in 0.2% Nonidet P-40 in PBS for 10 min. Cells were washed five times in PBS, and then blocked for 1 hour (hr) in 10% nonfat dry milk in PBS. After, cells were incubated in α-V5 mouse monoclonal antibodies (mAb) (Thermo Life Technologies) diluted 1:500 in 1% milk in PBS for 2 hr at RT. The cells were washed five times in PBS and incubated in goat anti-mouse IgG (H+L) cross-adsorbed secondary antibody, Alexa Fluor 488 (Thermo Fisher Scientific, catalog number A11001) 1:1000 in 1% milk in PBS for 2 hr at RT. Cells were washed five times in PBS and stained in 10 μg/mL of 4′,6-diamidino-2-phenylindole (DAPI) diluted in PBS for 15 min at RT. Cells were washed four times in PBS, twice in water, and then mounted onto microscope glass slides (Fisher Scientific) using a mounting medium (Southern Biotech). Cells' images were acquired using a Nikon E800 Upright fluorescence microscope (Nikon). *Western blotting:* Western analyses were performed as previously described (*Cestari, 2019*). Briefly, cleared lysates of *T. brucei* PFs were prepared in 1% Triton X-100 in PBS with 1 X protease inhibitor cocktail (Bio-Vision) and mixed in 4 X Laemmli buffer (Bio-Rad) with 710 mM β-mercaptoethanol and heated for 5 min at 95 °C. Proteins were resolved in 10% SDS/PAGE and transferred to nitrocellulose membranes (Sigma Aldrich). Membranes were probed for 2 h at RT (or overnight at 4 °C) with mAb α-V5 (BioShop Canada Inc, catalog number TAG006.100) 1:2500 in 6% milk in PBS 0.05% Tween (PBS-T). Membranes were incubated in 1:5,000 goat anti-mouse IgG (H+L)-HRP (Bio-Rad, catalog number 1706516) in 6% milk in PBS-T, washed in PBS-T, and developed by chemiluminescence using Supersignal West Pico Chemiluminescent Substrate (Thermo scientific). Images were acquired on a ChemiDoc MP imaging system (Bio-Rad). Membranes were stripped in 125 mM glycine, pH 2.0, and 1% SDS for 30 min, washed in PBS-T, and re-probed with mAb anti-mitochondrial heat shock protein (HSP) 70 of *T. brucei* (gift from Ken Stuart, Center for Global Infectious Diseases Research, Seattle Children's) 1:25 in 3% milk in PBS-T, followed by 1:5,000 goat anti-mouse IgG (H+L)-HRP (Bio-Rad, catalog number 1706516) in 3% milk in PBS-T and developed as described above.

## Protein expression and purification

The *T. brucei* RAP1 (Tb927.11.370) recombinant protein was expressed and purified as described before (*Cestari et al., 2019*). The RAP1 fragments rRAP1$^{1-300}$-His (aa 1–300), rRAP1$^{301-560}$-His (aa 301–560), or rRAP1$^{561-855}$-His (aa 561–855) were amplified by PCR (see *Supplementary file 6* for primers) from the *T. brucei* 427 genome and cloned into the pET-29a(+) vector (Novagen) using *NdeI* and *XhoI* restriction sites for the expression of proteins with a C-terminal 6xHis tag. Proteins were expressed and purified from 2 to 4 liters of *Escherichia coli* NiCo21(DE3) (New England Biolabs) as previously described (*Cestari et al., 2019*). Briefly, lysates were sonicated in PBS supplemented with 5 mM dithiothreitol (DTT), 0.2 mg/mL lysozyme, 0.05% NP-40, 10% glycerol, and 1 mM PMSF. For rRAP1$^{1-300}$-His, this buffer was supplemented with 7 M of urea (Fisher Scientific). After sonication, His-tagged proteins in the cleared lysate were purified using Profinity IMAC Nickel Charged Resin (Bio-Rad) and eluted in 50 mM sodium phosphate buffer with 300 mM NaCl pH 8 and 300 mM imidazole (Fischer Scientific). Eluted proteins were dialyzed overnight at 4 °C in binding buffer (25 mM HEPES, 150 mM NaCl, 10% glycerol, pH 7.5).

## Phosphoinositide binding assays

Phosphoinositide binding assays were performed as previously described (*Briggs et al., 2018*). Briefly, 1 μg of His-tagged recombinant protein (rRAP1-His, rRAP1$^{1-300}$-His, rRAP1$^{301-560}$-His, or rRAP1$^{561-855}$-His) was incubated with 1 μM of biotin-conjugated dioctanoylglycerol (diC8) PI(3,4,5)P3, diC8 PI(4,5)P2, InsP(1,4,5)P3 or InsP(1,3,4,5)P4 (Echelon Biosciences) rotating at RT for 1 hr. Afterward, magnetic streptavidin beads (Sigma Aldrich) (previously blocked overnight at 4 °C with 3% BSA in PBS) were

added and incubated at 4 °C for 1 hr rotating. Using a magnetic stand, the mix was washed 5–6 times in 25 mM HEPES pH 7.5, 300 mM NaCl, 0.2% NP-40 and 0.1% Tween 20, and then eluted in 50 µL of 2 x Laemmli buffer supplemented with 710 mM β-mercaptoethanol. Samples were boiled at 95 °C for 5 min. Samples were resolved in 10% SDS-PAGE, transferred onto a nitrocellulose membrane (Sigma Aldrich), and probed with α-His mAb (Thermo Scientific) diluted 1:1000 in 6% milk in PBS-T followed by α-mouse IgG HRP (Bio-Rad) diluted 1:5000 in 6% milk in PBS-T. Membranes developed as indicated above.

## Electrophoretic mobility shift assays

Synthetic single-stranded 5'-biotin-conjugated DNA sequences (*Supplementary file 6*) of telomeric repeats $(TTAGGG)_{10}$, 70 bp repeats (one repeat), and scrambled DNA (derived from telomeric repeats) were annealed (1:1 v/v, 10 µM) to reverse complementary unlabeled sequences (*Supplementary file 6*) (Integrated DNA Technologies) in a thermocycler by incubation at 95 °C for 10 min, 94 °C for 50 s, and a gradual decrease of 1 °C every 50 s for 72 cycles in 200 mM Tris–HCl pH 7.4, 20 mM $MgCl_2$, 500 mM NaCl. A total of 100 nM of annealed DNA were mixed with 1 µg of recombinant protein, and 2 mg/mL yeast tRNA (Life Technologies) in 20 mM HEPES, 40 mM KCl, 10 mM $MgCl_2$, 10 mM $CaCl_2$, and 0.2% N P-40, and incubated on a thermomixer (Eppendorf) at 37 °C for 1 hr. Then, 10 µL of the reaction was resolved on a 6% native gel (6% acrylamide in 200 mM Tris, 12,5 mM ethylenedi-aminetetraacetic acid (EDTA), pH 7.8 adjusted with acetic acid) at 100 V in 0.5 X TAE buffer (40 mM Tris, 20 mM acetic acid, and 1 mM EDTA). DNAs were transferred onto a nylon membrane (Life Technologies) at 100 V for 30 min in 0.5 x TAE. The membrane was blocked for 1 hr at RT with nucleic acid detection blocking buffer (Life Technologies), then probed for 1 hr at RT with streptavidin-HRP 1:5000 (Genescript) diluted in PBS 3% BSA. The membrane was washed in PBS-T and developed using Supersignal West Pico Chemiluminescent Substrate (Thermo Fisher Scientific). Images were acquired with a ChemiDoc MP imaging system (Bio-Rad).

## Microscale thermophoresis binding kinetics

Binding assays were performed using synthetic single-stranded 5'-Cy5 conjugated DNA sequences of telomeric repeats (TTAGGG)10, 70 bp repeats (one repeat), or scrambled DNA (derived from telomeric repeats) (see *Supplementary file 6* for sequences). Sequences were annealed (1:1 v/v, 10 µM) to reverse complementary unlabeled sequences (*Supplementary file 6*, all sequences synthesized by Integrated DNA Technologies) in a thermocycler by incubation at 95 °C for 10 min, 94 °C for 50 s, and a gradual decrease of 1 °C every 50 s for 72 cycles in 200 mM Tris–HCl pH 7.4, 20 mM $MgCl_2$, 500 mM NaCl. Then, 1 µM rRAP1-His was diluted in 16 twofold serial dilutions in 250 mM HEPES pH 7.4, 25 mM $MgCl_2$, 500 mM NaCl, and 0.25% (v/v) N P-40 and incubated with 20 nM telomeric or 70 bp repeats for 2 hr at 37 °C, gently shaking. Afterward, samples were centrifuged at 2000 x*g* for 5 min, loaded into Monolith capillary tubes (NanoTemper Technologies), and analyzed using the Monolith NT.115 MiscroScale Thermophoresis instrument with MO.Control software (NanoTemper Technologies). For binding assays in the presence of phosphatidylinositol phosphates (PIPs), the binding reaction was prepared as above but in the presence of 30 µM of diC8 PI(3,4,5)P3 or diC8 PI(4,5)P2. Four to six replicates were performed and presented as mean ± standard deviation. Data were analyzed using MO.Affinity Analysis (NanoTemper Technologies).

## Differential scanning fluorimetry

*T. brucei* rRAP1-His at 10 µM was incubated with 10 mM SYPRO Orange dye (ThermoFisher Scientific), filtered before use with a 0.22 µm sterile filter (Fisher scientific), in 15 mM HEPES, 150 mM NaCl, pH 7.0, and 10, 20, 30, or 40 µM of diC8 PI(3,4,5)P3, diC8 PI(4,5)P2, or no PIPs in 96-well microplates (Life Technologies). The plate was sealed with MicroAmp Optical Adhesive Film (Life Technologies), and the reaction mixture was placed on ice for 1 hr. Afterward, 20 µL of the reaction was added to a differential scanning fluorimetry plate (Life Technologies) and run using the Applied Biosystems StepOne-Plus instrument (ThermoFisher Scientific) according to the manufacturer's instructions. Briefly, samples were run using continuous ramp mode with two thermal profiles: Step 1 at 25 °C for 2 min and step 2 at 99 °C for 2 min, with a ramp rate of 100% at step 1 and 1% at step 2. Experiments were performed in triplicate and presented as mean ± standard deviation. Data analysis was performed using the

DSFworld (https://gestwickilab.shinyapps.io/dsfworld/) platform, and melting temperatures were calculated as the midpoint of the resulting fluorescence versus temperature curve.

## RNA-seq and real-time PCR

Poly-A enriched RNAs were extracted from $1.0 \times 10^8$ *T. brucei* BFs CN PIP5Pase exclusively expressing V5-tagged PIP5Pase WT or mutant D360A/N362A for 24 hr using the magnetic mRNA isolation kit (New England Biolabs) according to manufacturer's instructions. For procyclic forms, total RNAs were extracted from $5.0 \times 10^8$ *T. brucei* CN PIP5Pase growing in Tet + (0.5 µg/mL, no knockdown) or Tet – (knockdown) at 5 hr, 11 hr, 24 hr, 48 hr, and 72 hr using TRIzol (Thermo Fisher Scientific) according to manufacturer's instructions. The isolated mRNA samples were used to synthesize cDNA using Proto-Script II Reverse Transcriptase (New England Biolabs) according to the manufacturer's instructions. Real-time PCRs were performed using VSG primers as previously described (*Cestari and Stuart, 2015*). For RNA-seq, cDNA samples were purified using Totalpure NGS mag-bind (Omega-BioTek) at a 1.0 x ratio (beads to cDNA volume). cDNA samples were eluted in 55 µL of water, then fragmented to a size range of 300 bp – 1 kb (average 700 bp) using an M220 Focused-ultrasonicator (Covaris) with 75 peak incidence power, 10% duty factor, and 200 cycles per burst for 35 s at 20 °C. The fragmented cDNAs were then used for RNA-seq library preparation for Oxford nanopore sequencing (indicated below). Fifty fmol of barcoded libraries were sequenced in a MinION (Oxford Nanopore Technologies) using an FLO-MIN106 flow cell (Oxford Nanopore Technologies). Three biological replicates were performed.

## ChIP-seq

ChIP-seq was performed with *T. brucei* BFs CN PIP5Pase expressing endogenously HA-tagged RAP1 and exclusively expressing V5-tagged PIP5Pase WT or mutant D360A/N362A. Cells were grown for 24 hr in the absence of tet, which results in the exclusive expression of WT or mutant PIP5Pase, in HMI-9 media supplemented with 10% FBS, 2 µg/mL of G418, 2.5 µg/mL of phleomycin, 0.1 µg/mL of puromycin, and 25 µg/mL of nourseothricin. A total of $3.0 \times 10^8$ cells at mid-log growth were fixed in 1% paraformaldehyde at 4 °C for 10 min, then quenched for 10 min in 140 mM glycine. Fixed cells were spun down at 2000 x*g* for 10 min, washed twice in PBS, and processed for ChIP using truChIP Chromatin Shearing Kit with Formaldehyde (Covaris) according to the manufacturer's instructions. DNA was sheared to a size range of 0.5–1.5 kb (average 700 bp) using an M220 Focused-ultrasonicator (Covaris) with 75 peak incidence power, 5% duty factor, and 200 cycles per burst for 7 min at 7 °C. ChIP was performed with a total of 15 µg of sonicated chromatin and 10 µg of Mab anti-V5 (BioShop Canada Inc). The antibodies were cross-linked to Protein G Mag Sepharose Xtra (GE Healthcare) according to the manufacturer's instructions before immunoprecipitations. Antibody eluted chromatin was incubated with 20 units of Proteinase K (Thermo Fisher Scientific) and reverse cross-linked at 65 °C overnight, followed by DNA extraction by phenol:chloroform:isoamyl alcohol (25:24:1, ThermoFisher Scientific) and isopropanol (v/v) precipitation. Samples were resuspended in water, and the DNA was selected for fragments above 300 bp using Totalpure NGS mag-bind (Omega-BioTek) at 0.85 x ratio (beads to DNA volume). The DNA was eluted in water and used to prepare Oxford nanopore DNA sequencing libraries (indicated below). Fifty fmol of barcoded libraries was sequenced in a MinION (Oxford Nanopore Technologies) using an FLO-MIN106 flow cell (Oxford Nanopore Technologies). Three biological replicates were performed.

## VSG-seq

*T. brucei* BFs PIP5Pase CN cell lines exclusively expressing V5-tagged PIP5Pase WT or mutant D360A/N362A were seeded at $1.0 \times 10^3$ parasites/mL in 50 mL of HMI-9 media supplemented with 10% FBS, 2 µg/mL of G418, 2.5 µg/mL of phleomycin, and grown for 24 hr in the absence of tet, which results in the exclusive expression of WT or PIP5Pase mutant. Then, tet (0.5 µg/mL) was added to Mut PIP5Pase cells to rescue WT PIP5Pase expression, and cells were grown for an additional 60 hr at 37 °C and 5% $CO_2$ until they reached a concentration of $1.0 \times 10^6$ parasites/mL. RNA was then isolated using Trizol (Sigma-Aldrich) according to the manufacturer's instructions. cDNA was synthesized using random primers with 5 x Protoscript II Reverse Transcriptase (New England BioLabs Ltd). cDNA was amplified using the VSG Splice Leader and SP6-VSG14mer primers (*Mugnier et al., 2015*; *Supplementary file 6*) with denaturing at 94 °C for 3 min, followed by 22 cycles of 94 °C for 1 min, 42 °C for 1 min,

and 72 °C for 2 min (*Mugnier et al., 2015*). Amplified fragments were purified using 0.55 x beads/sample ratio with Mag-Bind TotalPure NGS (Omega Bio-Tek). Amplicons were prepared for Oxford Nanopore sequencing using the ligation sequencing kit SQK-LSK110 (Oxford Nanopore Technologies) and PCR Barcoding Expansion kit (EXP-PBC001) according to the manufacturer's instructions (described below). Five fmol pooled barcoded libraries were sequenced in a MinION using a Flongle FLO-FLG001 flow cell (Oxford Nanopore Technologies). Experiments were performed in three biological replicates.

## Clonal-VSG-seq

*T. brucei* BFs PIP5Pase CN cell lines seeded at $1.0x10^4$ parasites/mL in 10 mL of HMI-9 media supplemented with 10% FBS and 2 µg/mL of G418 and 2.5 µg/mL of phleomycin in the absence of tet (tet -) for PIP5Pase knockdown, or in the presence of tet (tet +, 0.5 µg/mL) for PIP5Pase expression. After 24 hr, tet (0.5 µg/mL) was added to the tet – cells to restore PIP5Pase expression and cells were cloned by limited dilution. A protocol for clonal VSG switching is available (*Touray et al., 2023*). Briefly, the cells were diluted to a concentration of 1 parasite/300 µL and grown in 200 µL of media in 96-well culture plates (Thermo Scientific Inc) and grown for 5–7 days at 37 °C and 5% $CO_2$. Clones were collected from the plates and transferred to 24-well culture plates (Bio Basic Inc) containing 2 mL of HMI-9 media with 10% FBS, 2 µg/mL of G418, 2.5 µg/mL of phleomycin, and 0.5 µg/mL of tet for 24 hr to reach $1.0x10^6$ parasites/mL. Afterward, cells were harvested by centrifugation at 2,000 x*g*, and pellets were collected for RNA extraction using the 96 Well Plate Bacterial Total RNA Miniprep Super Kit (Bio Basic Inc) according to the manufacturer's instructions. cDNA was synthesized from extracted RNA samples using M-MuLV reverse transcriptase kit (New England Biolabs Ltd) based on the manufacturer's instructions and using customized primers splice leader (SL_F) and VSG barcoded primers (BCA_Rd_3'AllVSGs 1–96) (see *Supplementary file 6* for primer sequences). The forward SL F primer is complementary to the splice leader sequence and has an Oxford nanopore barcode adapter sequence. The reverse primer contains a sequence complementary to the 3'-region conserved among VSG genes, a unique 6 random nucleotide barcode sequence, and an Oxford nanopore barcode adapter sequence for library preparation. The synthesized cDNAs were pooled in a 1.5 mL Eppendorf tube and amplified using the Oxford nanopore library barcoding primers (PCR Barcoding Expansion kit EXP-PBC001, Oxford Nanopore Technologies) with denaturing at 95° for 3 min, followed by 22 cycles of 95 °C for 30 s, 60 °C for 30 s, and 72 °C for 2.30 min. Amplified fragments were purified using 0.55 x beads/sample ratio NucleoMag NGS magnetic beads (Takara Bio). Amplicons were prepared for Oxford Nanopore sequencing using the ligation sequencing kit SQK-LSK110 (Oxford Nanopore Technologies) according to the manufacturer's instructions (details described below). Five fmol of the barcoded library were sequenced in a MinION (Oxford Nanopore Technologies) using a Flongle FLO-FLG001 flow cell (Oxford Nanopore Technologies). A total of 118 and 94 clones of CN PIP5Pase cells tet + (PIP5Pase expression) and tet – (PIP5Pase knockdown), respectively, were analyzed.

## Preparation of DNA libraries and Oxford nanopore sequencing

DNA libraries were prepared using the ligation sequencing kit SQK-LSK110 (Oxford Nanopore Technologies), according to the manufacturer's instructions. Briefly, the fragmented nucleic acids were end-repaired and A-tailed using the NEBNext Ultra II End Repair and A-Tailing Module (New England Biolabs). The samples were cleaned with Totalpure NGS mag-bind (Omega-BioTek) at 0.85 x ratio (beads to DNA volume), then ligated with nanopore barcode adapters. The product was subsequently used for PCR amplification with barcoding primers (Oxford Nanopore Technologies). The PCR product was cleaned with Totalpure NGS mag-bind (Omega-BioTek) at a 0.85 x ratio (beads to DNA volume) and ligated to motor proteins for DNA sequencing. The libraries were sequenced in a MinION Mk1C sequencer (Oxford Nanopore Technologies) using FLO-MIN106 flow cells (unless otherwise stated), and sequences were basecalled using the Guppy software (Oxford Nanopore Technologies). Sequencing information is available in *Supplementary file 5* and fastq data is available in the Sequence Read Archive (SRA) with the BioProject identification PRJNA934938.

## Computational analysis of RNA-seq, VSG-seq and ChIP-seq

RNA-seq or ChIP-seq fastq data from nanopore sequencing were mapped to *T. brucei* 427–2018 reference genome using minimap2 (https://github.com/lh3/minimap2; *Li, 2023*; *Li, 2018*) and the

alignment data were processed with SAMtools (https://github.com/samtools/samtools; *Danecek et al., 2021*; *samtools, 2023*). Reads with a nanopore sequencing Q-score >7 were used for analysis, and the mean Q-score was 12. RNA-seq and VSG-seq (including clonal VSG-seq) mapped reads were quantified using featureCounts from package Subread and used for differential expression analysis using EgdeR. Alignments were filtered to remove supplementary alignments using samtools flags. Because RNA-seq reads were detected mapping to subtelomeric regions in cells expressing Mut PIP5Pase, differential expression analysis was also performed with alignments filtered for primary reads with a stringent mapping probability of 99,9% (mapQ 30) to eliminate potential multiple mapping reads. Read counts were obtained with featureCounts counting primary alignments only and processed using EgdeR. For ChIP-seq, aligned reads mapped with minimap2 were processed with deepTools (https://deeptools.readthedocs.io/en/develop/) for coverage analysis using bamCoverage and enrichment analysis using bamCompare comparing RAP1-HA ChIP vs Input. Peak calling and statistical analysis were performed for broad peaks with Model-based Analysis of ChIP-Seq MACS3 (https://github.com/macs3-project/MACS, *Zhang et al., 2008*; *MACS3 project team, 2023*) and data were visualized using the integrated genomics viewer tool (Broad Institute). To identify if RAP1-HA was mapping specifically to silent vs active ESs, mapped reads were filtered to remove supplementary and secondary alignments, that is only primary alignments were considered. Analyses were also performed with a minimum mapping probability of 90% and 99% (mapQ 10 and 20, see *Supplementary file 4*). This stringent analysis removes reads aligning to multiple regions of the genome including reads mapping to multiple ESs, that is it retains reads aligning to specific regions of the ESs. Filtered reads were processed as described above using deepTools and MACS3 analysis. Scripts used for ChIP-seq, RNA-seq, and VSG-seq analysis are available at https://github.com/cestari-lab/lab_scripts (copy archived at *Cestari, 2023a*). A specific pipeline was developed for clonal VSG-seq analysis, available at https://github.com/cestari-lab/VSG-Bar-seq (copy archived at *Cestari, 2023b*; *Touray et al., 2023*).

## Quantification of PI(3,4,5)P3

*T. brucei* BFs PIP5Pase CN cell lines exclusively expressing V5-tagged PIP5Pase WT or mutant D360A/N362A were grown in 100 mL HMI-9 media supplemented with 10% FBS, 2 µg/mL of G418, 2.5 µg/mL of phleomycin, and grown for 24 hr in the absence of tet, which results in the exclusive expression of WT or Mut PIP5Pase. A total of $1.0 \times 10^8$ cells were used for PI(3,4,5)P3 quantification using Echelon's PIP3 Mass ELISA Kit (Echelon Biosciences), according to the manufacturer's instructions. Briefly, the cells were centrifuged at 2000 x*g* at RT for 5 min, washed in PBS 6 mM Glucose, and the supernatant was discarded. The cell pellets were resuspended in 10 mL of ice-cold 0.5 M Trichloroacetic Acid (TCA), incubated on ice for 5 min, then centrifuged at 1000 x*g* for 7 min at 4 °C. The supernatant was discarded, and the pellets were washed twice using 3 mL of 5% TCA/1 mM EDTA per wash at RT. Neutral lipids were extracted by adding 3 mL of MeOH: CHCl$_3$ (2:1) to the pellets and vortexed for 10 min at RT, followed by centrifugation at 1000 x*g* for 5 min, and the supernatant discarded. This step was repeated once more to remove neutral lipids. 2.25 mL of MeOH:CHCl$_3$:HCl (80:40:1) was added to the pellets and vortexed for 25 min at RT, then centrifuged at 1000 x*g* for 5 min to extract the acidic lipids. The supernatants were transferred to new 15 mL centrifuge tubes. 0.75 mL CHCl$_3$ and 1.35 mL 0.1 N HCl were added to the supernatants, vortexed for 30 s, then centrifuged at 1000 x*g* for 5 min to separate the organic and aqueous phases. 1.5 mL of the organic (lower) phase was transferred to new 2 mL vials and vacuum dried using a SpeedVac (Thermo Scientific) at RT. The dried lipids were stored at –20 °C until used. To quantify PI(3,4,5)P3 interacting with RAP1-HA, 300 mL of *T. brucei* BFs CN PIP5Pase expressing endogenously HA-tagged RAP1 and exclusively expressing V5-tagged PIP5Pase WT or mutant D360A/N362A were grown for 24 hr in absence of tet, which results in the exclusive expression of WT or mutant PIP5Pase, in HMI-9 media supplemented with 10% FBS, 2 µg/mL of G418, 2.5 µg/mL of phleomycin, 0.1 µg/mL of puromycin, and 25 µg/mL of nourseothricin. A total of $3 \times 10^8$ cells (PIP5Pase WT and Mut) were centrifuged at 2,000 x*g* at RT for 5 min, washed in 20 mL of PBS 6 mM glucose and then lysed in 5 mL of *T. brucei* cell lysis buffer (50 mM Tris, 150 mM NaCl, 2 X protein inhibitor cocktail, 0.1% NP-40, and 1% Triton-X100), rotating at 4 °C for 30 min. Lysates were centrifuged at 15,000 x*g*, 4 °C for 30 min, and the cleared lysate was transferred to a new 15 mL falcon tube. Two percent (100 µL) of the cleared lysate was collected for Western blot of HA-tagged RAP1 (input). The remaining cleared lysate was used for RAP1-HA immunoprecipitation with anti-HA

monoclonal antibodies (ABClonal). Twenty μg of anti-HA monoclonal antibodies crosslinked to protein G MicroBeads (GE Healthcare) were added to the cleared lysate and incubated overnight at 4 °C. Immunoprecipitated samples on beads were washed five times in wash buffer (50 mM Tris, 150 mM NaCl, 2 X protein inhibitor cocktail, 0.1% NP-40). An aliquote corresponding to 20% of the beads-bound samples was collected and eluted in 6 M Urea/100 mM Glycine pH 2.9 for western blot analysis of immunoprecipitated RAP1-HA protein. The remaining samples on the beads (80%) were used for acidic lipids extraction, as indicated above. The dried lipids were resuspended in PBS-T 0.25% Protein Stabilizer (PBS-T 0.25% PS), and PI (3,4,5)P3 content was measured using Echelon's PIP3 Mass ELISA Kit (Echelon Biosciences), according to the manufacturer's instructions.

## Data presentation and statistical analysis

Data are shown as means ± SEM from at least three biological replicates. Comparisons among groups were made by a two-tailed t-test using GraphPad Prism. *P*-values <0.05 with a confidence interval of 95% were considered statistically significant. Graphs were prepared using Prism (GraphPad Software, Inc), MATLAB (Mathworks), or Integrated Genome Viewer (Broad Institute).

## Acknowledgements

This research was enabled in part by computational resources provided by Calcul Quebec (https://www.calculquebec.ca/en/) and the Digital Research Alliance of Canada (alliancecan.ca). Funding: Canadian Institutes of Health Research grant CIHR PJT-175222 (IC); The Natural Sciences and Engineering Research Council of Canada grant RGPIN-2019–05271 (IC); Fonds de Recherche du Québec - Nature et Technologie grant 2021-NC-288072 (IC); Canada Foundation for Innovation grant JELF 258389 (IC); McGill University fund 130251 (IC); Islamic Development Bank Scholarship 600042744 (AOT); FRQNT-Ukraine postdoctoral fellowship BUKX:2022–2023 337989 (OK); The Natural Sciences and Engineering Research Council of Canada CGS M fellowship (ML).

## Additional information

### Funding

| Funder | Grant reference number | Author |
| --- | --- | --- |
| Canadian Institutes of Health Research | CIHR PJT-175222 | Igor Cestari |
| Natural Sciences and Engineering Research Council of Canada | RGPIN-2019–05271 | Igor Cestari |
| Fonds de Recherche du Québec - Nature et technologies | 2021-NC-288072 | Igor Cestari |
| Canada Foundation for Innovation | JELF 258389 | Igor Cestari |
| McGill University | 130251 | Igor Cestari |
| Islamic Development Bank Scholarship | 600042744 | Abdoulie O Touray |
| FRQNT-Ukraine postdoctoral fellowship | BUKX:2022–2023 337989 | Oksana Kutova |
| Natural Sciences and Engineering Research Council of Canada | CGS M fellowship | Mira Loock |

The funders had no role in study design, data collection and interpretation, or the decision to submit the work for publication.

## Author contributions
Abdoulie O Touray, Validation, Investigation, Visualization, Methodology, Writing – original draft, Writing – review and editing; Rishi Rajesh, Tony Isebe, Mira Loock, Formal analysis, Investigation, Methodology, Writing – review and editing; Tamara Sternlieb, Formal analysis, Investigation, Visualization, Methodology, Writing – review and editing; Oksana Kutova, Formal analysis, Writing – review and editing; Igor Cestari, Conceptualization, Resources, Data curation, Software, Formal analysis, Supervision, Funding acquisition, Validation, Investigation, Visualization, Methodology, Writing – original draft, Project administration, Writing – review and editing

## Author ORCIDs
Abdoulie O Touray (ID) https://orcid.org/0000-0002-2983-1559
Tony Isebe (ID) http://orcid.org/0000-0003-3094-1832
Tamara Sternlieb (ID) http://orcid.org/0000-0002-1167-5367
Mira Loock (ID) http://orcid.org/0009-0000-4738-6463
Igor Cestari (ID) http://orcid.org/0000-0003-3845-7535

Reviewer #1 (Public Review): https://doi.org/10.7554/eLife.89331.4.sa1
Reviewer #2 (Public Review): https://doi.org/10.7554/eLife.89331.4.sa2
Author Response https://doi.org/10.7554/eLife.89331.4.sa3

---

# Additional files

## Supplementary files
• Supplementary file 1. RNA-seq analysis of *T. brucei* bloodstream form cells expressing mutant vs wildtype PIP5Pase. Gene expression analysis by RNA-seq using Oxford nanopore sequencing of *T. brucei* bloodstream forms exclusively expressing mutant (D360A/N362A) compared to wildtype PIP5Pase. cDNA was synthesized from poly-A enriched RNAs. The data combines three biological replicates. LogFC, log2 fold-change; LogCPM, log2 counts per million; FDR, false discovery rate, calculated using the Benjamini-Hochberg method.

• Supplementary file 2. Statistical analysis of RAP1-HA ChIP-seq in *T. brucei* expressing WT PIP5Pase. Location of ChIP-seq peaks and statistical analysis were obtained using Model-based Analysis of ChIP-Seq (MACS3) set for broad peaks search. The *q*-values (false discovery rate) are adjusted *p*-values by the Benjamini-Hochberg method. Fold enrichment is calculated for the region against random Poisson distribution with background lambda. Pileup indicates the mean value across all values in the entire peak region. Data for BESs, MES, and centromeres were included in *Supplementary file 4*, hence not included in this dataset. Three biological replicates comparing RAP1-HA ChIP versus Input were included in the analysis.

• Supplementary file 3. VSGs identified in VSG-seq experiments (related to *Figure 1E*). Data compares VSGs expressed in cells expressing WT (wild type) or Mut (Mutant, D360A/N362A) PIP5Pase. Mutant PIP5Pase was temporarily knocked down for 24 h followed by re-expression for 60 h. Values are counts per million in log2.

• Supplementary file 4. Analysis of RAP1-HA binding sites on BESs, MES, and centromeres by ChiP-seq comparing ChIP vs Input. Location of peaks and statistical analysis were obtained using Model-based Analysis of ChIP-Seq (MACS3). The *q*-values (false discovery rate) are adjusted *p*-values by the Benjamini-Hochberg method. The location of centromeres was obtained by analysis of ChIP-seq data from the kinetoplastid kinetochore protein 2 (KKT2) generated by *Akiyoshi and Gull, 2014*; accession numbers SRX372731 and SRX372732 mapped to the *T. brucei* 427–2018 genome reference. FE, fold enrichment for the region against random Poisson distribution with background lambda; Pileup, mean value across all values in the entire peak region.

• Supplementary file 5. Oxford nanopore sequencing metrics for RNA-seq and ChIP-seq. RNA-seq in cells expressing WT or Mut (D360A/N362A) PIP5Pase, and ChIP-seq of RAP1-HA in cells expressing WT or Mut PIP5Pase.

• Supplementary file 6. List of oligonucleotides used in this study. Oligonucleotides used for cloning RAP1 recombinant DNA, synthesis of telomeric and 70 bp repeats, synthesis of CRISPR guide RNAs, and clonal VSG-seq library preparation. BCA_Rd, nanopore barcode adaptor and random 6 nucleotide sequence. Oligonucleotide modifications, e.g., 5'-biotin (5Biosg) or 5'-Cy5 fluorescence modification (5Cy5) are indicated. Oligonucleotides are shown from 5' to 3'.

- MDAR checklist

## Data availability

Sequencing information is available in Supplementary File 5 and fastq data is available in the Sequence Read Archive (SRA) with the BioProject identification PRJNA934938.

The following dataset was generated:

| Author(s) | Year | Dataset title | Dataset URL | Database and Identifier |
|---|---|---|---|---|
| Touray AO, Rajesh R, Isebe T, Sternlieb T, Loock M, Kutova O, Cestari I | 2023 | *Trypanosoma brucei* brucei strain:Lister 427 | https://www.ncbi.nlm. nih.gov/bioproject/ PRJNA934938 | NCBI BioProject, PRJNA934938 |

The following previously published dataset was used:

| Author(s) | Year | Dataset title | Dataset URL | Database and Identifier |
|---|---|---|---|---|
| Akiyoshi B, Gull K | 2014 | ChIP-seq of *Trypanosoma brucei* bloodstream form: eYFP::KKT2 immunoprecipitated DNA | https://www.ncbi. nlm.nih.gov/sra/ SRX13241378 | NCBI Sequence Read Archive, SRX13241378 |

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
