## [Editor Report · eLife assessment]

*Trypanosoma brucei* evades mammalian humoral immunity through the expression of different variant surface glycoprotein genes. In this **fundamental** paper, the authors extend previous observations that TbRAP1 both interacts with PIP5Pase and binds PI(3,4,5)P3, indicating a role for PI(3,4,5)P3 binding. They therefore suggest that antigen switching might have a signal-dependent component. The evidence is mostly **compelling**, but with some caveats because tagged proteins were used.

---

## [Referee Report · Reviewer #1 (Public Review)]

*Trypanosoma brucei* undergoes antigenic variation to evade the mammalian host's immune response. To achieve this, *T. brucei* regularly expresses different VSGs as its major surface antigen. VSG expression sites are exclusively subtelomeric, and VSG transcription by RNA polymerase I is strictly monoallelic. It has been shown that *T. brucei* RAP1, a telomeric protein, and the phosphoinositol pathway are essential for VSG monoallelic expression. In previous studies, Cestari et al. (ref. 24) has shown that PIP5pase interacts with RAP1 and that RAP1 binds PI(3,4,5)P3. RNAseq and ChIPseq analyses have been performed previously in PIP5pase conditional knockout cells, too (ref. 24). In the current study, Touray et al. did similar analyses except that catalytic dead PIP5pase mutant was used and the DNA and PI(3,4,5)P3 binding activities of RAP1 fragments were examined. Specifically, the authors examined the transcriptome profile and did RAP1 ChIPseq in PIP5pase catalytic dead mutant. The authors also expressed several C-terminal His6-tagged RAP1 recombinant proteins (full-length, aa1-300, aa301-560, and aa 561-855). These fragments' DNA binding activities were examined by EMSA analysis and their phosphoinositides binding activities were examined by affinity pulldown of biotin-conjugated phosphoinositides. As a result, the authors confirmed that VSG silencing (both BES-linked and MES-linked VSGs) depends on PIP5pase catalytic activity, but the overall knowledge improvement is incremental. The most convincing data come from the phosphoinositide binding assay as it clearly shows that N-terminus of RAP1 binds PI(3,4,5)P3 but not PI(4,5)P2, although this is only assayed in vitro, while the in vivo binding of full-length RAP1 to PI(3,4,5)P3 has been previously published by Cestari et al (ref. 24) already. Considering that many phosphoinositides exert their regulatory role by modulating the subcellular localization of their bound proteins, it is reasonable to hypothesize that binding to PI(3,4,5)P3 can remove RAP1 from the chromatin.

---

## [Referee Report · Reviewer #2 (Public Review)]

In this manuscript, Touray et al investigate the mechanisms by which PIP5Pase and RAP1 control VSG expression in *T. brucei* and demonstrate an important role for this enzyme in a signalling pathway that likely plays a role in antigenic variation in *T. brucei*. While these data do not definitively show a role for this pathway in antigenic variation, the data are critical for establishing this pathway as a potential way the parasite could control antigenic variation and thus represent a fundamental discovery.

The methods used in the study are generally well-controlled. The authors provide evidence that RAP1 binds to PI(3,4,5)P3 through its N-terminus and that this binding regulates RAP1 binding to VSG expression sites, which in turn regulates VSG silencing. Overall their results support the conclusions made in the manuscript. Readers should take into consideration that the epitope tags on RAP1 could alter its function, however.

There are a few small caveats that are worth noting. First, the analysis of VSG derepression and switching in Figure 1 relies on a genome which does not contain minichromosomal (MC) VSG sequences. This means that MC VSGs could theoretically be mis-assigned as coming from another genomic location in the absence of an MC reference. As the origin of the VSGs in these clones isn't a major point in the paper, I do not think this is a major concern, but I would not over-interpret the particular details of switching outcomes in these experiments.

Another aspect of this work that is perhaps important, but not discussed much by the authors, is the fact that signalling is extremely poorly understood in *T. brucei*. In Figure 1B, the RNA-seq data show many genes upregulated after expression of the Mut PIP5Pase (not just VSGs). The authors rightly avoid claiming that this pathway is exclusive to VSGs, but I wonder if these data could provide insight into the other biological processes that might be controlled by this signaling pathway in .

Overall, this is an excellent study which represents an important step forward in understanding how antigenic variation is controlled in *T. brucei*. The possibility that this process could be controlled via a signalling pathway has been speculated for a long time, and this study provides the first mechanistic evidence for that possibility.

---

## [Author Response]

The following is the authors’ response to the previous reviews

**Reviewer #1 (Public Review):**
Comments on the original submission:*Trypanosoma brucei* undergoes antigenic variation to evade the mammalian host's immune response. To achieve this, *T. brucei* regularly expresses different VSGs as its major surface antigen. VSG expression sites are exclusively subtelomeric, and VSG transcription by RNA polymerase I is strictly monoallelic. It has been shown that *T. brucei* RAP1, a telomeric protein, and the phosphoinositol pathway are essential for VSG monoallelic expression. In previous studies, Cestari et al. (ref. 24) has shown that PIP5pase interacts with RAP1 and that RAP1 binds PI(3,4,5)P3. RNAseq and ChIPseq analyses have been performed previously in PIP5pase conditional knockout cells, too (ref. 24). In the current study, Touray et al. did similar analyses except that catalytic dead PIP5pase mutant was used and the DNA and PI(3,4,5)P3 binding activities of RAP1 fragments were examined. Specifically, the authors examined the transcriptome profile and did RAP1 ChIPseq in PIP5pase catalytic dead mutant. The authors also expressed several C-terminal His6-tagged RAP1 recombinant proteins (full-length, aa1300, aa301-560, and aa 561-855). These fragments' DNA binding activities were examined by EMSA analysis and their phosphoinositides binding activities were examined by affinity pulldown of biotin-conjugated phosphoinositides. As a result, the authors confirmed that VSG silencing (both BES-linked and MES-linked VSGs) depends on PIP5pase catalytic activity, but the overall knowledge improvement is incremental. The most convincing data come from the phosphoinositide binding assay as it clearly shows that N-terminus of RAP1 binds PI(3,4,5)P3 but not PI(4,5)P2, although this is only assayed in vitro, while the in vivo binding of full-length RAP1 to PI(3,4,5)P3 has been previously published by Cestari et al (ref. 24) already. Considering that many phosphoinositides exert their regulatory role by modulate the subcellular localization of their bound proteins, it is reasonable to hypothesize that binding to PI(3,4,5)P3 can remove RAP1 from the chromatin. However, no convincing data have been shown to support the author's hypothesis that this regulation is through an "allosteric switch".Comments on revised manuscript:In this revised manuscript, Touray et al. have responded to reviewers' comments with some revisions satisfactorily. However, the authors still haven't addressed some key scientific rigor issues, which are listed below:1. It is critical to clearly state whether the observations are made for the endogenous WT protein or the tagged protein. It is good that the authors currently clearly indicate the results observed in vivo are for the RAP1-HA protein. However, this is not as clearly stated for in vitro EMSA analyses. In addition, in discussion, the authors simply assumed that the c-terminally tagged RAP1 behaves the same as WT RAP1 and all conclusions were made about WT RAP1.There are two choices here. The authors can validate that RAP1-HA still retains RAP1's essential function as a sole allele in *T. brucei* cells (as was recommended previously). Indeed, HA-tagged RAP1 has been studied before, but it is the N-terminally HA-tagged RAP1 that has been shown to retain its essential functions. Adding the HA tag to the C-terminus of RAP1 may well cause certain defects to RAP1. For example, N-terminally HA-tagged TERT does not complement the telomere shortening phenotype in TERT null *T. brucei* cells, while C-terminally GFP-tagged TERT does, indicating that HA-TERT is not fully functional while TERT-GFP likely has its essential functions (Dreesen, RU thesis). Although RAP1-HA behaves similar to WT RAP1 in many ways, it is still not fully validated that this protein retains essential functions of RAP1. By the way, it has been published that cells lacking one allele of RAP1 behave as WT cells for cell growth and VSG silencing (Yang et al. 2009, Cell; Afrin et al. 2020, mSphere). In addition, although RAP1 may interact with TRF weakly, the interaction is direct, as shown in yeast 2-hybrid analysis in (Yang et al. 2009, Cell).Alternatively, if the authors do not wish to validate the functionality of RAP1-HA, they need to add one paragraph at the beginning of the discussion to clearly state that RAP1-HA may not behave exactly as WT RAP1. This is important for readers to better interpret the results. In addition, the authors need to tune down the current conclusions dramatically, as all described observations are made on RAP1-HA but not the WT RAP1.

The results with RAP1-HA are consistent with previous knowledge of RAP1 interactions with telomeric proteins and DNA. Hence, the C-terminal HA-tagged RAP1 seems, by all measures, functional. Nevertheless, to make it clear for the reader, we added a note in the discussion, lines 244-246: “Although we showed that C-terminal HA-tagged RAP1 protein has telomeric localization (Cestari et al. 2015, PNAS) and interactions with other telomeric proteins (Cestari et al. 2019 Mol Cell Biol); we cannot rule out potential differences between HA-tagged and non tagged RAP1.”

For a similar reason, the current EMSA results truly reflect how C-terminally His6-tagged RAP1 and RAP1 fragments behave. If the authors choose not to remove the His6 tag, it is essential that they use "RAP1-His6" to refer to these recombinant proteins. It is also essential for the authors to clearly state in the discussion that the tagged RAP1 fragments bind DNA, but the current data do not reveal whether WT RAP1 binds DNA. In addition, the authors incorrectly stated that "disruption of the MybL domain sequence did not eliminate RAP1-telomere binding in vivo" (lines 165-166). In ref 29, deletion of Myb domain did not abolish RAP1-telomere association. However, point mutations in MybL domain that abolish RAP1's DNA binding activities clearly disrupted RAP1's association with the telomere chromatin. Therefore, the current observation is not completely consistent with that published in ref 29.

We stated in line 149-150 “…we expressed and purified from *E. coli* recombinant 6xHistagged *T. brucei* RAP1 (rRAP1)”. To clarify to the authors, we replaced rRAP1 with rRAP1-His throughout the manuscript and figures. As for the statement that “disruption of the MybL domain sequence did not eliminate RAP1-telomere binding in vivo" (lines 165-166).”. We removed the statement from the manuscript.

1. There is no evidence, in vitro or in vivo, that binding PI(3,4,5)P3 to RAP1 causes conformational change in RAP1. The BRCT domain of RAP1 is known for its ability to homodimerize (Afrin et al. 2020, mSphere). It is therefore possible that binding of PI(3,4,5)P3 to RAP1 simply disrupts its homodimerization function. The authors clearly have extrapolated their conclusions based on available data. It is therefore important to revise the discussion and make appropriate statements.

We did not state that PI(3,4,5)P3 causes RAP1 conformational changes. We discussed the possibility. We stated in lines 199-201: “PI(3,4,5)P3 inhibition of RAP1-DNA binding might be due to its association with RAP1 N-terminus causing conformational changes that affect Myb and MybL domains association with DNA.” This is a reasonable discussion, given the data presented in the manuscript.

**Reviewer #2 (Public Review):**
In this manuscript, Touray et al investigate the mechanisms by which PIP5Pase and RAP1 control VSG expression in *T. brucei* and demonstrate an important role for this enzyme in a signalling pathway that likely plays a role in antigenic variation in *T. brucei*. While these data do not definitively show a role for this pathway in antigenic variation, the data are critical for establishing this pathway as a potential way the parasite could control antigenic variation and thus represent a fundamental discovery.The methods used in the study are generally well-controlled. The authors provide evidence that RAP1 binds to PI(3,4,5)P3 through its N-terminus and that this binding regulates RAP1 binding to VSG expression sites, which in turn regulates VSG silencing. Overall their results support the conclusions made in the manuscript. Readers should take into consideration that the epitope tags on RAP1 could alter its function, however.There are a few small caveats that are worth noting. First, the analysis of VSG derepression and switching in Figure 1 relies on a genome which does not contain minichromosomal (MC) VSG sequences. This means that MC VSGs could theoretically be mis-assigned as coming from another genomic location in the absence of an MC reference. As the origin of the VSGs in these clones isn't a major point in the paper, I do not think this is a major concern, but I would not over-interpret the particular details of switching outcomes in these experiments.

We agree with the reviewer and thus made no speculations on minichromosomes. The data analysis must rely on the available genome, and the reference genome used is well-assembled with PacBio sequences and Hi-C data (Muller et al. 2018, Nature).

Another aspect of this work that is perhaps important, but not discussed much by the authors, is the fact that signalling is extremely poorly understood in *T. brucei*. In Figure 1B, the RNA-seq data show many genes upregulated after expression of the Mut PIP5Pase (not just VSGs). The authors rightly avoid claiming that this pathway is exclusive to VSGs, but I wonder if these data could provide insight into the other biological processes that might be controlled by this signaling pathway in *T. brucei*.

We published that the inositol phosphate pathway also plays a role in *T. brucei* development (Cestari et al. 2018, Mol Biol Cell; reviewed by Cestari I 2020, PLOS Pathogens)

Overall, this is an excellent study which represents an important step forward in understanding how antigenic variation is controlled in *T. brucei*. The possibility that this process could be controlled via a signalling pathway has been speculated for a long time, and this study provides the first mechanistic evidence for that possibility.
**Reviewer #1 (Recommendations For The Authors):**
Please see the public review for recommendations.1. It is critical to clearly state whether the observations are made for the endogenous WT protein or the tagged protein. It is good that the authors currently clearly indicate the results observed in vivo are for the RAP1-HA protein. However, this is not as clearly stated for in vitro EMSA analyses. In addition, in discussion, the authors simply assumed that the c-terminally tagged RAP1 behaves the same as WT RAP1 and all conclusions were made about WT RAP1.There are two choices here. The authors can validate that RAP1-HA still retains RAP1's essential function as a sole allele in *T. brucei* cells (as was recommended previously). Indeed, HA-tagged RAP1 has been studied before, but it is the N-terminally HA-tagged RAP1 that has been shown to retain its essential functions. Adding the HA tag to the C-terminus of RAP1 may well cause certain defects to RAP1. For example, N-terminally HA-tagged TERT does not complement the telomere shortening phenotype in TERT null *T. brucei* cells, while C-terminally GFP-tagged TERT does, indicating that HA-TERT is not fully functional while TERT-GFP likely has its essential functions (Dreesen, RU thesis). Although RAP1-HA behaves similar to WT RAP1 in many ways, it is still not fully validated that this protein retains essential functions of RAP1. By the way, it has been published that cells lacking one allele of RAP1 behaves as WT cells for cell growth and VSG silencing (Yang et al. 2009, Cell; Afrin et al. 2020, mSphere). In addition, although RAP1 may interact with TRF weakly, the interaction is direct, as shown in yeast 2-hybrid analysis in (Yang et al. 2009, Cell).Alternatively, if the authors do not wish to validate the functionality of RAP1-HA, they need to add one paragraph at the beginning of the discussion to clearly state that RAP1-HA may not behave exactly as WT RAP1. This is important for readers to better interpret the results. In addition, the authors need to tune down the current conclusions dramatically, as all described observations are made on RAP1-HA but not the WT RAP1.

The results with RAP1-HA are consistent with previous knowledge of RAP1 interactions with telomeric proteins and DNA. Hence, the C-terminal HA-tagged RAP1 seems, by all measures, functional. Nevertheless, to make it clear for the reader, we added a note in the discussion, lines 244-246: “Although we showed that C-terminal HA-tagged RAP1 protein has telomeric localization (Cestari et al. 2015, PNAS) and interactions with other telomeric proteins (Cestari et al. 2019 Mol Cell Biol); we cannot rule out potential differences between HA-tagged and non tagged RAP1.”

For a similar reason, the current EMSA results truly reflect how C-terminally His6-tagged RAP1 and RAP1 fragments behave. If the authors choose not to remove the His6 tag, it is essential that they use "RAP1-His6" to refer to these recombinant proteins. It is also essential for the authors to clearly state in the discussion that the tagged RAP1 fragments bind DNA, but the current data do not reveal whether WT RAP1 binds DNA. In addition, the authors incorrectly stated that "disruption of the MybL domain sequence did not eliminate RAP1-telomere binding in vivo" (lines 165-166). In ref 29, deletion of Myb domain did not abolish RAP1-telomere association. However, point mutations in MybL domain that abolish RAP1's DNA binding activities clearly disrupted RAP1's association with the telomere chromatin. Therefore, the current observation is not completely consistent with that published in ref 29.

We stated in lines 149-150 “…we expressed and purified from *E. coli* recombinant 6xHistagged *T. brucei* RAP1 (rRAP1)”. To clarify to the authors, we replaced rRAP1 with rRAP1-His throughout the manuscript text. As for the statement that “disruption of the MybL domain sequence did not eliminate RAP1telomere binding in vivo" (lines 165-166).”. We removed the statement from the manuscript.

1. There is no evidence, in vitro or in vivo, that binding PI(3,4,5)P3 to RAP1 causes conformational change in RAP1. The BRCT domain of RAP1 is known for its ability to homodimerize (Afrin et al. 2020, mSphere). It is therefore possible that binding of PI(3,4,5)P3 to RAP1 simply disrupts its homodimerization function. The authors clearly have extrapolated their conclusions based on available data. It is therefore important to revise the discussion and make appropriate statements.

We did not state that PI(3,4,5)P3 causes RAP1 conformational changes. We discussed the possibility. We stated in lines 199-201: “PI(3,4,5)P3 inhibition of RAP1-DNA binding might be due to its association with RAP1 N-terminus causing conformational changes that affect Myb and MybL domains association with DNA.” This is a reasonable discussion, given the data presented in the manuscript.